# Calcium transfer from the ER to other organelles for optimal signaling in *Toxoplasma gondii*

Zhu-Hong Li[1], Beejan Asady[1,2], Le Chang[1], Myriam Andrea Hortua Triana[1], Catherine Li[1], Isabelle Coppens[2], Silvia NJ Moreno[1]*

[1]Center for Tropical and Emerging Global Diseases, University of Georgia and Department of Cellular Biology, University of Georgia, Athens, United States; [2]Department of Molecular Microbiology and Immunology, Johns Hopkins Bloomberg School of Public Health, Baltimore, United States

*For correspondence: smoreno@uga.edu

**Competing interest:** The authors declare that no competing interests exist.

## eLife Assessment

This **important** study shows that calcium stores in the endoplasmic reticulum of the parasitic protozoan, *Toxoplasma gondii* play a major role in regulating calcium levels in the cytosol as well as other organelles such as the mitochondrion. Advanced imaging techniques, including use of genetically encoded calcium indicators provide **compelling** evidence for the role of the SERCA-$Ca^{2+}$-ATPase pump in regulating organellar calcium levels. However, it remains unclear whether inter-organellar calcium transport occurs via ER-mitochondria membrane contact sites or other mechanisms. This work will be of interest to cell and molecular biologists interested in calcium signalling in divergent eukaryotes.

**Abstract** $Ca^{2+}$ signaling in cells begins with the opening of $Ca^{2+}$ channels in either the plasma membrane (PM) or endoplasmic reticulum (ER), leading to a sharp increase in the physiologically low (<100 nM) cytosolic $Ca^{2+}$ level. The temporal and spatial regulation of $Ca^{2+}$ is crucial for the precise activation of key biological processes. In the apicomplexan parasite *Toxoplasma gondii*, which infects approximately one-third of the global population, $Ca^{2+}$ signaling governs essential aspects of the parasite's infection cycle. *T. gondii* relies on $Ca^{2+}$ signals to regulate pathogenic traits, with several $Ca^{2+}$-signaling components playing critical roles. $Ca^{2+}$ entry from the extracellular environment has been demonstrated in *T. gondii* for both, extracellular parasites, exposed to high $Ca^{2+}$, and intracellular parasites, which acquire $Ca^{2+}$ from host cells during host $Ca^{2+}$ signaling events. Active egress, an essential step of the parasite's infection cycle, is preceded by a large increase in cytosolic $Ca^{2+}$, most likely initiated by release from intracellular stores. However, extracellular $Ca^{2+}$ is also necessary to reach a cytosolic $Ca^{2+}$ threshold required for timely egress. In this study, we investigated the mechanism of intracellular $Ca^{2+}$ store replenishment and identified a central role for the SERCA-$Ca^{2+}$-ATPase in maintaining $Ca^{2+}$ homeostasis within the ER and in other organelles. We demonstrate mitochondrial $Ca^{2+}$ uptake, which occurs by transfer of $Ca^{2+}$ from the ER, likely through membrane contact sites. Our findings suggest that the *T. gondii* ER plays a key role in sequestering and redistributing $Ca^{2+}$ to intracellular organelles following $Ca^{2+}$ influx at the PM.

## Introduction

*Toxoplasma gondii* is an intracellular parasite from the Apicomplexan Phylum that infects approximately one-third of the world population (*Weiss and Dubey, 2009*). During the initial infection, *T.

*gondii* undergoes multiple rounds of a lytic cycle, which consists of host cell invasion, replication within a parasitophorous vacuole (PV), exit from the host cell causing its lysis followed by reinvasion of new host cells (**Black and Boothroyd, 2000**; **Blader et al., 2015**). Cytosolic $Ca^{2+}$ ($[Ca^{2+}]_c$) fluctuations precede the activation of several key steps of the *T. gondii* lytic cycle like motility, attachment, invasion, and egress (**Lourido and Moreno, 2015**; **Hortua Triana et al., 2018**). Egress from the host cell is an essential step for the infection cycle of *T. gondii* (**Bisio and Soldati-Favre, 2019**) and it was shown that it is preceded by a cytosolic $Ca^{2+}$ increase (**Endo et al., 1982**; **Borges-Pereira et al., 2015**). Extracellular $Ca^{2+}$ entry was demonstrated in extracellular (**Pace et al., 2014**; **Hortua Triana et al., 2024**) and intracellular replicating tachyzoites (**Vella et al., 2021**). This activity was highly regulated, and work from our lab revealed that a TRP-like channel activity was involved (**Márquez-Nogueras et al., 2021**).

$Ca^{2+}$ signaling is part of the signaling pathways that regulate a large number of cellular functions (**Clapham, 2007**). All cells express a variety of channels, transporters, and $Ca^{2+}$ pumps, located at the PM and/or intracellular organelles (ER, acidic stores, and mitochondria) that regulate/control the concentration of cytosolic $Ca^{2+}$. However, an elevated cytosolic $Ca^{2+}$ concentration sustained for prolonged periods is toxic to cells and may result in their death (**Bootman and Bultynck, 2020**).

In *T. gondii*, both $Ca^{2+}$ entry through the plasma membrane and release from intracellular stores like the ER may initiate a cascade of signaling events important for the stimulation of the biological steps of the parasite lytic cycle (**Lourido and Moreno, 2015**; **Hortua Triana et al., 2018**). $Ca^{2+}$ oscillations were observed in motile parasites loaded with fluorescent $Ca^{2+}$ indicators (**Lovett and Sibley, 2003**), as well as expressing Genetically Encoded Calcium Indicators (GECIs) (**Borges-Pereira et al., 2015**). The significance of $Ca^{2+}$ signals during all stages of the lytic cycle has been demonstrated, but little is known about the mechanism by which intracellular stores contribute to cytosolic $Ca^{2+}$ signals and downstream regulation.

The ER, an organelle unique to eukaryotic cells, is the main store for $Ca^{2+}$ in most eukaryotes. It has been proposed that the ER is functionally heterogeneous, with $Ca^{2+}$-binding proteins, pumps and channels, distributed nonuniformly, resulting in the presence of distinct subdomains within the organelle (**Papp et al., 2003**). The ER in mammalian cells facilitates $Ca^{2+}$ tunneling through its lumen as a mechanism of delivering $Ca^{2+}$ to targeted sites without activating inappropriate processes in the cell cytosol (**Petersen et al., 2017**). In addition, the ER is ubiquitously distributed and is in close contact with all cellular organelles and the PM (**Spang, 2018**). Over the past decade, a new paradigm has emerged that seeks to decipher how subcellular organelles communicate with each other in order to coordinate activities and efficiently distribute ions and lipids within the cell. Numerous observations have highlighted the presence of tight, stable, and yet non-fusogenic associations between organellar membranes which have since become known as membrane contact sites (MCSs) (**Phillips and Voeltz, 2016**).

The secretory pathway of *T. gondii* is organized in a highly polarized manner with the ER being an extension of the nuclear envelope (**Hager et al., 1999**; **Tomavo et al., 2013**). The ER at the apical surface of the nuclear envelope is continuous with the Golgi stacks and extends toward the secretory organelles, micronemes and rhoptries, which are unique to the apicomplexan phylum (**Hager et al., 1999**). These organelles perform important functions required for a successful lytic cycle, including host cell attachment, invasion, and establishment of the parasitophorous vacuole (PV). Cytoplasmic $Ca^{2+}$ increases, due to release from the ER, have been reported to initiate responses like microneme secretion (**Carruthers and Sibley, 1999**), conoid extrusion (**Del Carmen et al., 2009**), invasion (**Vieira and Moreno, 2000**; **Lovett and Sibley, 2003**) and egress (**Arrizabalaga and Boothroyd, 2004**). These responses require precise spatiotemporal control of diverse targets and suggests the presence of distinct systems to deliver $Ca^{2+}$ to specific locations rather than allowing global increases, which would activate unnecessary and potentially detrimental signaling events (**Huet and Moreno, 2023**).

In order to concentrate $Ca^{2+}$ ions, the ER utilizes a SERCA-$Ca^{2+}$-ATPase, a transmembrane P-type ATPase, that couples ATP hydrolysis to the transport of ions across biological membranes and against a concentration gradient. SERCA pumps can be inhibited by various inhibitors, including the very potent and highly specific thapsigargin (TG) (**Thastrup et al., 1990**; **Sagara and Inesi, 1991**).

*T. gondii* expresses a SERCA $Ca^{2+}$-ATPase (TgSERCA), which possesses conserved SERCA domains, $Ca^{2+}$-binding sites, and residues required for ATP hydrolysis (**Nagamune et al., 2007a**). The function of TgSERCA was determined by rescue experiments of yeast cells defective in $Ca^{2+}$-ATPases and by

its specific inhibition by TG (*Nagamune et al., 2007a*). TgSERCA was mainly localized to the ER of *T. gondii* but also showed a distinct distribution in extracellular parasites, where the protein was partially found in ER vesicles in the apical region near micronemes (*Nagamune et al., 2007a*). This distribution pattern was different from the one obtained with the transient transfection of GFP-HDEL (an ER marker), which was retained near the nuclear envelope, suggesting an uneven distribution of ER markers in extracellular parasites. The authors suggested that this distribution to the apical end may be important for rapid release and effective recovery of cytosolic $Ca^{2+}$, events that likely govern both motility and microneme secretion (*Nagamune et al., 2007a*; *Nagamune et al., 2007b*).

In this work, we investigate the role of the ER in intracellular calcium handling in *Toxoplasma gondii*. Specifically, we explore how the ER contributes to calcium uptake following extracellular influx and how it facilitates calcium redistribution to other organelles. Using genetic and pharmacological tools, we examine the activity of the SERCA $Ca^{2+}$-ATPase and its role in coordinating calcium dynamics. Our findings support a model in which the ER acts as a central hub for calcium buffering and transfer, driven by the high calcium affinity of TgSERCA.

## Results

### The ER sequesters and redistributes $Ca^{2+}$ to other organelles following PM influx

We initially designed experiments to track the destination of $Ca^{2+}$ taken up by extracellular tachyzoites from the extracellular milieu into their cytosol. Tachyzoites were loaded with the ratiometric $Ca^{2+}$ indicator Fura-2 (*Figure 1A*) and incubated in Ringer buffer containing 100 µM EGTA to chelate extracellular $Ca^{2+}$ and prevent further uptake. Under these conditions, we investigated the release of intracellular $Ca^{2+}$ pools by various pharmacological agents.

When TG, an inhibitor of SERCA, was added, it blocked the re-uptake of $Ca^{2+}$ into the ER and unmasked the passive leak of $Ca^{2+}$ from the ER into the cytosol, resulting in a cytosolic $Ca^{2+}$ increase of approximately 160 nM (*Figure 1B–C*). We compared this response to the effect of ionomycin (IO), a $Ca^{2+}/H^+$ ionophore which acts on neutral intracellular $Ca^{2+}$ stores, inducing depletion of $Ca^{2+}$, mainly from the ER (*Smith et al., 1989*). Exposure of *T. gondii* to 1 µM IO caused a cytosolic increase of approximately 700–1,100 nM $Ca^{2+}$ (*Figure 1C*). We next tested another SERCA inhibitor, cyclopiazonic acid (CPA), which is structurally unrelated to TG and with a different mode of action (*Inesi and Sagara, 1994*). CPA induced a smaller increase in cytosolic $Ca^{2+}$ compared to TG, possibly indicating less efficacy toward TgSERCA (*Figure 1—figure supplement 1A–B*). Exposure to TG before CPA abolished the effect of CPA, whereas exposure to CPA did not prevent the effect of TG. (*Figure 1—figure supplement 1B–C*). Although TG does a better job at depleting the ER of $Ca^{2+}$ in intact parasites, the resulting increase of cytosolic $Ca^{2+}$ after adding TG is modest compared to the response of IO (*Figure 1C*). This result could reflect the slow kinetics of $Ca^{2+}$ leak from the ER, allowing other buffering and transport mechanisms to mitigate the phenomenon. Alternatively, it may indicate that the duration after TG treatment was sufficient to complete store depletion. As shown in *Figure 1B–C*, residual $Ca^{2+}$ remains in the stores after TG treatment, and the TG-induced phenomenon does not return to baseline, suggesting that the leak remains active.

Next, we tested the effect of $Ca^{2+}$ entry in the filling of intracellular stores by measuring the cytosolic $Ca^{2+}$ increases in response to inhibitors after pre-exposing the parasite suspension to extracellular $Ca^{2+}$ (*Figure 1D–G*, *darker traces*). Addition of 1.8 mM $Ca^{2+}$ caused a cytosolic increase due to influx through the plasma membrane (*Figure 1D*, *dark blue trace*) (*Pace et al., 2014*). A substantial portion of the entering $Ca^{2+}$ appeared to be rapidly sequestered by the ER, as evidenced by the significantly greater TG-induced response in parasites previously exposed to extracellular $Ca^{2+}$ (compare light and dark blue traces) (*Figure 1D*). We next tested the lysosomotropic agent glycyl-L-phenylalanine-naphthylamide (GPN), which primarily mobilizes $Ca^{2+}$ from acidic organelles (*Haller et al., 1996*; *Lloyd-Evans et al., 2008*; *Miranda et al., 2010*; *Yuan et al., 2021*) and observed a similar pattern (*Figure 1E*). The increase in cytosolic $Ca^{2+}$ following GPN addition was markedly greater in parasites previously exposed to extracellular $Ca^{2+}$ than in those that had not been exposed (*Figure 1E*, *compare light and dark blue traces*). We previously proposed that GPN may act on the lysosome-like Plant-Like Vacuolar Compartment (PLVAC), a dynamic acidic organelle involved in calcium storage, and the processing of secretory proteins (*Miranda et al., 2010*; *Stasic et al., 2022*). The increase in cytosolic

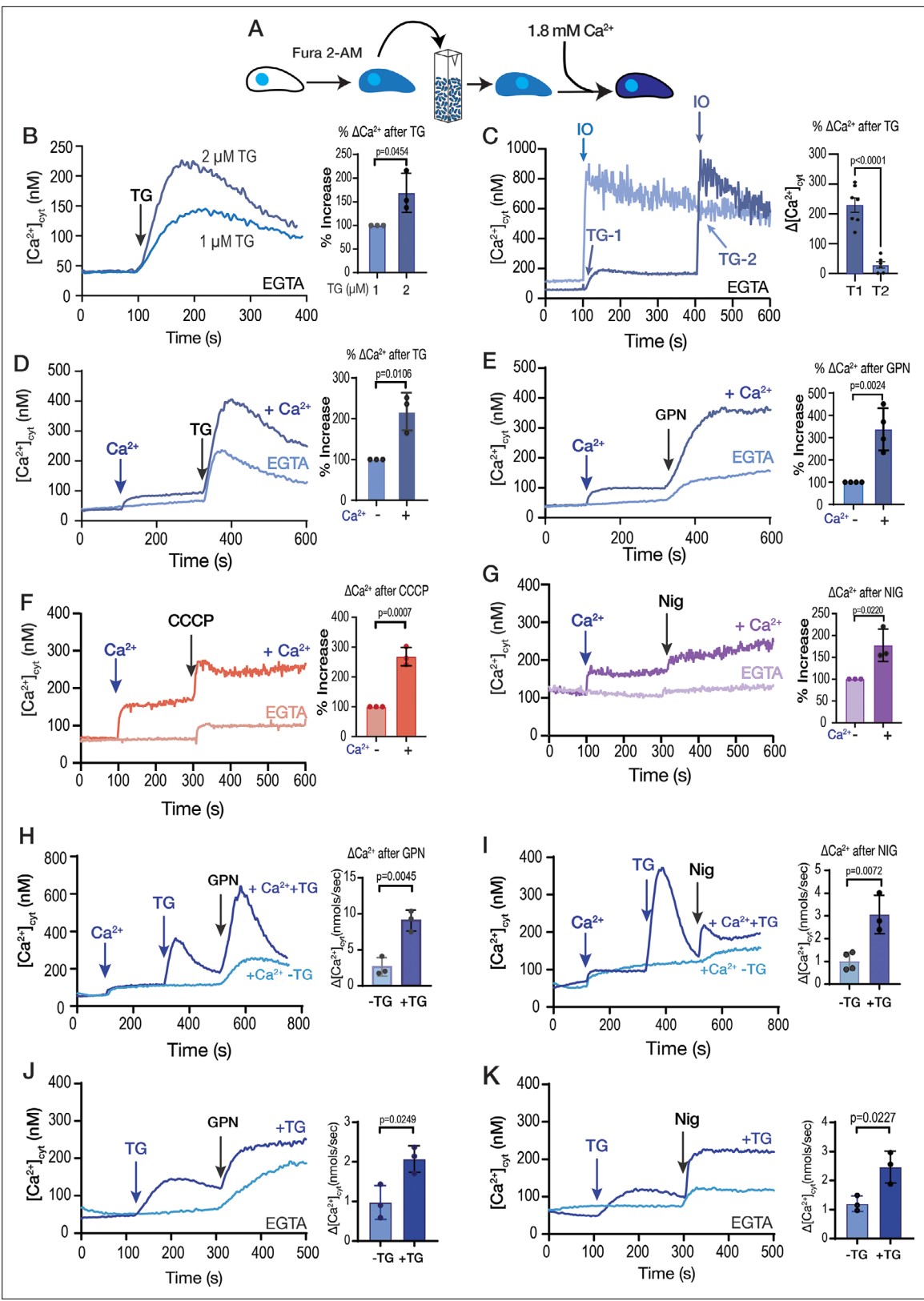

**Figure 1.** The role of extracellular calcium in the filling of intracellular stores. (**A**) Scheme depicting the Fura-2-AM loading and the experimental setup. (**B**) *T. gondii* tachyzoites loaded with Fura-2 were in suspension in Ringer buffer with 100 µM EGTA. Thapsigargin (TG) was added at 100 s, at two different concentrations (1 and 2 µM). (**C**) Same conditions as in B. 1 µM Thapsigargin (TG) was added at 100 s for T1 and 400 s for T2. 1 µM ionomycin (IO) was added at 400 sec for trace 1 (T1) and at 100 sec for trace 2 (T2). Bar graph shows Ca²⁺ increase after adding TG before (T1) and after

*Figure 1 continued on next page*

*Figure 1 continued*

IO (T2). (**D**) Same conditions as in B. 1.8 mM CaCl$_2$ was added at 100 s, followed by 2 µM TG at 300 s (dark blue trace). The light blue trace shows the same experiment without the addition of CaCl$_2$. (**E**) Similar to D but using 40 µM glycyl-L-phenylalanine-naphthylamide (GPN) instead of TG. (**F**) Same experimental setup to the one shown in D but using the mitochondrial uncoupler CCCP. (**G**) Same experimental setup to the one shown in D but adding the potassium ionophore nigericin (Nig), 10 µM. The quantification for D, E, F, and G shows the % increase of cytoplasmic calcium compared with the same condition without previous addition of calcium. (**H**) 1.8 mM CaCl$_2$ was added at 100 s, 1 µM TG was added at 300 sec followed by 40 µM GPN at 500 s. The light blue trace shows the same experiment without the addition of TG. The quantification shows the Ca$^{2+}$ increase after adding GPN ± previous addition of TG. (**I**) Identical experiment to H but instead using 10 µM Nig at 500 sec. The quantification shows the Ca$^{2+}$ increase after adding Nig ± previous addition of TG. (**J**) 1 µM TG was added at 100 s followed by 40 µM GPN at 300 s. The light blue trace shows the same experiment without the addition of TG. The buffer contains 100 µM EGTA. Quantification shows Ca$^{2+}$ increase after adding GPN ± previous addition of TG. (**K**) Similar conditions to J but with 10 µM Nig at 300 s instead. Data are presented as mean ± SD for all comparisons. *p*-value: unpaired two-tailed t-test performed in all comparisons.

The online version of this article includes the following source data and figure supplement(s) for figure 1:

**Source data 1.** Source data for *Figure 1* showing Fura2 calcium measurements.

**Figure supplement 1.** Intracellular calcium pools.

**Figure supplement 2.** Intracellular calcium pools.

Ca$^{2+}$ in response to the addition of nigericin (which acts on acidic stores) or CCCP (which likely targets mitochondria) was also greater in cells previously loaded with extracellular Ca$^{2+}$ (*Figure 1F–G*, *dark purple and dark orange traces, respectively*). These data indicate that the ER, mitochondrion, PLVAC, and other acidic stores release more calcium into the cytosol of *T. gondii* tachyzoites following exposure to extracellular Ca$^{2+}$, which stimulates its influx through the plasma membrane. The ER displayed high capacity to access a large portion of extracellular Ca$^{2+}$, with TG producing close to ~300–400 nM of Ca$^{2+}$ increase after pre-exposure to Ca$^{2+}$ and only ~150–200 nM Ca$^{2+}$ without Ca$^{2+}$ pre-exposure (*Figure 1D*).

We next aimed to understand how other compartments are replenished with Ca$^{2+}$, given that the ER appears to be particularly effective at taking up Ca$^{2+}$ from the cytosol. We designed an experiment where parasites were first loaded with Ca$^{2+}$, followed by inhibition of SERCA using TG. This inhibition prevents ER Ca$^{2+}$ uptake, allowing Ca$^{2+}$ to accumulate on the cytosolic side of the ER membrane. Under these conditions, we added agonists such as GPN or nigericin following the addition of TG. As shown in *Figure 1H*, Ca$^{2+}$ was first added to load intracellular stores, followed by TG to induce ER Ca$^{2+}$ leakage, and then GPN to trigger Ca$^{2+}$ release from acidic stores. Comparison of GPN-induced cytosolic Ca$^{2+}$ signals with and without TG pre-treatment revealed a significantly greater response in the TG condition. A similar enhancement was observed for the nigericin-induced response following TG treatment (*Figure 1I*).

These results support our hypothesis that extracellular Ca$^{2+}$ is primarily taken up by the ER and subsequently redistributed to other organelles. Importantly, Ca$^{2+}$ was added prior to TG to allow store loading, and TG treatment then permitted ER Ca$^{2+}$ leakage, facilitating Ca$^{2+}$ transfer to other compartments.

However, we considered the possibility that the enhanced responses to GPN or nigericin could be due to increased PM Ca$^{2+}$ influx triggered by elevated cytosolic Ca$^{2+}$. To test this, we repeated the experiments in the absence of extracellular Ca$^{2+}$ (*Figure 1J–K*). Notably, prior addition of TG resulted in an enhanced cytosolic Ca$^{2+}$ response to both GPN and nigericin. These results further support the notion that Ca$^{2+}$ can be transferred from the ER to other intracellular stores independently of extracellular Ca$^{2+}$ influx.

We also performed an additional experiment in which SERCA was inhibited with TG prior to Ca$^{2+}$ addition. We then quantified the subsequent GPN response in conditions with and without TG preincubation and observed a significant increase in the TG-treated group (*Figure 1—figure supplement 2A*). This result suggests that, under non-physiological conditions where SERCA is blocked, the PLVAC may take up Ca$^{2+}$ directly from the cytosol. However, this is unlikely to occur under normal conditions, as functional SERCA likely has a higher affinity for Ca$^{2+}$ and would sequester it limiting its availability to other compartments.

In summary, pre-exposure of *T. gondii* to physiological levels of extracellular Ca$^{2+}$ markedly enhanced the capacity of the ER, mitochondria, and acidic stores to release Ca$^{2+}$ into the cytosol, with the ER and GPN-sensitive stores exhibiting the most pronounced responses.

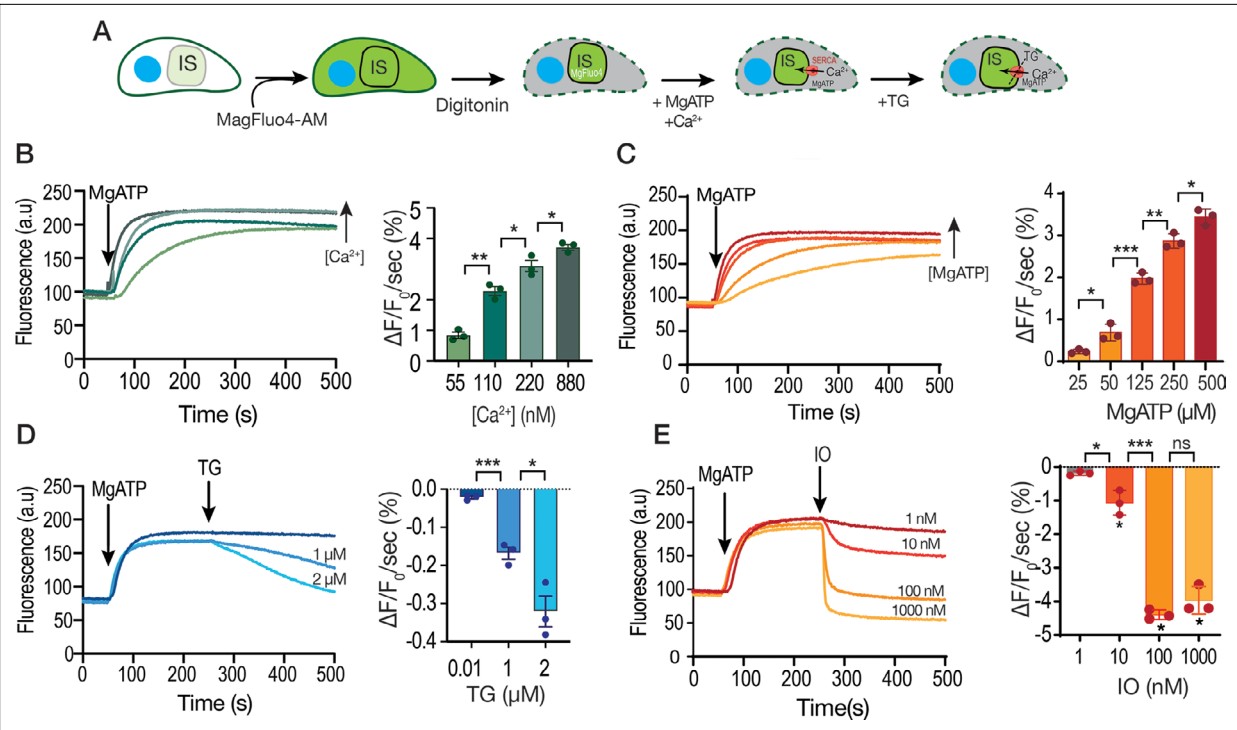

**Figure 2.** Ca²⁺ uptake by intracellular stores. (**A**) Scheme showing the loading with Mag-Fluo-4 AM followed by permeabilization with digitonin of a *T. gondii* tachyzoite (RH parental strain) suspension (IS, intracellular store). (**B**) Fluorescence measurements (see Materials and methods for specifics) of the suspension of parasites loaded with Mag-Fluo-4. MgATP (500 μM), the sarco/endoplasmic reticulum Ca²⁺-ATPase (SERCA) substrate was added at 50 s. The bar graph shows the quantification of the slope of the increase in fluorescence after adding MgATP. The concentration of free Ca²⁺ was varied, and it is indicated. The calculation of free Ca²⁺ was done using MaxChelator. (**C**) A similar experimental setup to the one shown in B with 220 nM free Ca²⁺, with varied concentrations of MgATP as indicated in the bar graph, which shows the quantification of the slope of fluorescence increase after adding MgATP. (**D**) Experiment was done with 500 μM MgATP and 220 nM free Ca²⁺. Thapsigargin (TG) was added to inhibit SERCA causing calcium to be released from the store. The concentrations used are indicated. The bar graph shows the negative slope after the addition of TG. (**E**) Similar to D, but adding various concentrations of ionomycin (IO). The concentrations used are indicated and the slopes were measured after the addition of IO. Data are presented as mean ± SD for B-E. *p*-value: unpaired two-tailed t-test performed for all comparisons. ns, not significant, *p*>0.05. *, *p*≤0.05. **, *p*≤0.01. ***, *p*≤0.001. ****, *p*≤0.0001.

The online version of this article includes the following source data and figure supplement(s) for figure 2:

**Source data 1.** Source data for *Figure 2* showing MagFluo4 calcium measurements.

**Figure supplement 1.** Inhibition of sarco/endoplasmic reticulum Ca²⁺-ATPase (SERCA) activity by thapsigargin (TG) and cyclopiazonic acid (CPA).

## Ca²⁺ uptake by the SERCA-Ca²⁺ ATPase in permeabilized tachyzoites

The previous results highlight the central role of the ER in taking up Ca²⁺ from the cytosol following an influx from the extracellular milieu. We propose that this ER uptake, essential for maintaining Ca²⁺ store levels, is driven by the high Ca²⁺ affinity of TgSERCA. As a key mechanism, SERCA enables the ER to sustain its Ca²⁺ concentration despite the constitutive and passive leakage of Ca²⁺ from the ER into the cytosol (*Camello et al., 2002*).

To characterize the activity of TgSERCA *in situ*, we adapted a protocol to directly measure Ca²⁺ uptake by the stores in which TgSERCA localizes (ER and Golgi apparatus) (*Calixto et al., 2025*). This approach, which has been widely used in mammalian cells to assess Ca²⁺ release from the ER, employs the low-affinity Ca²⁺ indicator Mag-Fluo-4 (Kd ~22 μM) (*Rossi and Taylor, 2020*). The cytosolic concentration of Ca²⁺ in *T. gondii* is approximately 70 nM (*Moreno and Zhong, 1996*), which is well below the detection threshold of Mag-Fluo-4. To facilitate loading into organelles, we incubated parasites for an extended period with higher concentrations of Mag-Fluo-4-AM, promoting its compartmentalization into intracellular stores. Following incubation, parasites were washed and treated with a low concentration of digitonin, which selectively permeabilizes the plasma membrane while preserving the integrity of organellar membranes. Under these conditions, the parasites retained the Ca²⁺ indicator within their organelles (*Figure 2A*). We next assessed the capacity of these permeabilized parasites to take

up $Ca^{2+}$. Since the activity of SERCA depends on MgATP (*Figure 2B*), we added this substrate in the presence of defined $Ca^{2+}$ concentrations calculated using the MaxChelator program (*Bers et al., 1994*). Under these conditions (free calcium ranging from 55 to 880 nM and MgATP at concentrations of 25–500 µM), we observed consistent and reproducible $Ca^{2+}$ uptake, as shown in *Figure 2B–C*. We selected 220 nM $Ca^{2+}$ for our study because this concentration approximates physiological cytosolic fluctuations and supports detectable $Ca^{2+}$ uptake. Additionally, this concentration of $Ca^{2+}$ has been used in previous studies of mammalian SERCA (*Rossi and Taylor, 2020*). Validation that this activity is mediated by TgSERCA is demonstrated by the addition of TG, which inhibits SERCA allowing $Ca^{2+}$ leakage from the organelle (*Figure 2D*). Although the $Ca^{2+}$ released after adding TG appears modest, consistent with the slow leak characteristics of ER calcium, the high Kd (22 µM) of the indicator implies that even small decreases in fluorescence signal, represent significant $Ca^{2+}$ efflux. In contrast, IO at 1 µM caused a more pronounced $Ca^{2+}$ release, lowering the $Ca^{2+}$ concentration below the baseline level. This effect is likely due to IO targeting multiple intracellular compartments in addition to the ER, as well as the fundamental difference in mechanisms: IO acts as an ionophore, directly facilitating $Ca^{2+}$ efflux across membranes, whereas TG inhibits SERCA, resulting in $Ca^{2+}$ release through the ER's natural leak pathway (*Figure 2E*).

We used the Mag-Fluo-4 assay to directly compare the inhibitory effects of CPA and TG (*Figure 2—figure supplement 1*). Under the conditions of the Mag-Fluo-4 assay, using digitonin-permeabilized parasites, both inhibitors produced comparable levels of $Ca^{2+}$ efflux suggesting that at the concentrations used both inhibited SERCA and the efflux rate corresponds to the intrinsic ER leak mechanism (*Figure 2—figure supplement 1A–C*). This finding suggests that CPA may be less effective at inhibiting SERCA in intact parasites, possibly due to its reversibility and partial dissociation over time, allowing residual $Ca^{2+}$ reuptake into the ER and resulting in a smaller cytosolic $Ca^{2+}$ increase compared to TG.

In summary, these results demonstrate that the activity of TgSERCA in *T. gondii* tachyzoites can be measured *in situ* using permeabilized parasites loaded with the low-affinity $Ca^{2+}$ indicator Mag-Fluo-4. This activity is MgATP-dependent and both TG and CPA can inhibit TgSERCA activity, leading to leakage of the accumulated $Ca^{2+}$. The larger effect of IO compared to TG is likely due to differences in their mechanisms of action.

## TgSERCA and the *T. gondii* lytic cycle

To investigate the role of TgSERCA (TGGT1_230420) in the biology of *T. gondii*, we generated conditional knockout parasites (*i△TgSERCA*), based on the gene's predicted essentiality (fitness score –5.44) (*Sidik et al., 2016*). A tetracycline regulatable element was inserted at the 5' end of the *TgSERCA* gene locus to control its expression with anhydrotetracycline (ATc) (*Sheiner et al., 2011*). In addition, we endogenously tagged TgSERCA with a C-terminal 3xHA epitope and generated clonal lines of both *i△TgSERCA* and *i△TgSERCA-3HA* (*Figure 3A*).

With the aim of detecting the protein, we generated a guinea pig polyclonal antibody against the phosphorylation (P) and nucleotide-binding (N) domains of TgSERCA, which was affinity-purified and validated by Western blotting (*Figure 3—figure supplement 1A*) and IFAs (*Figure 3—figure supplement 1B*). Colocalization of the anti-TgSERCA with the anti-HA signal was confirmed by IFA. Although the signals from the anti-HA and anti-TgSERCA did not completely overlap, both were lost in the *i△TgSERCA* mutant grown in the presence of ATc (*Figure 3—figure supplement 1B, +ATc*). The partial colocalization may reflect differences in antibody accessibility or that the two antibodies recognize distinct regions of the protein. Both Western blots and IFAs confirmed that TgSERCA expression is tightly regulated by ATc and becomes undetectable after 2.5 days in culture (*Figure 3B*, *Figure 3—figure supplement 1B*). Growth of the *i△TgSERCA* mutant was severely impaired in the presence of ATc, as assessed by plaque assays (*Figure 3C–D*). In this assay, parasites undergo successive rounds of invasion, replication, and egress, leading to host cell lysis and the formation of plaques on confluent monolayers. Downregulation of TgSERCA expression led to a marked defect in replication, with parasites failing to progress beyond one or two rounds of division (*Figure 3E–F*). All parasitophorous vacuoles (PVs) in ATc-treated cultures contained four or fewer parasites (*Figure 3F*, *bar graph on the right*). Host cell invasion was also reduced in the *i△TgSERCA* mutant (*Figure 3G*) when cultured with ATc.

Parasite egress was significantly affected by TgSERCA depletion. Ionomycin (IO), which has been known to trigger egress by inducing $Ca^{2+}$ release (*Borges-Pereira et al., 2015*), and natural egress

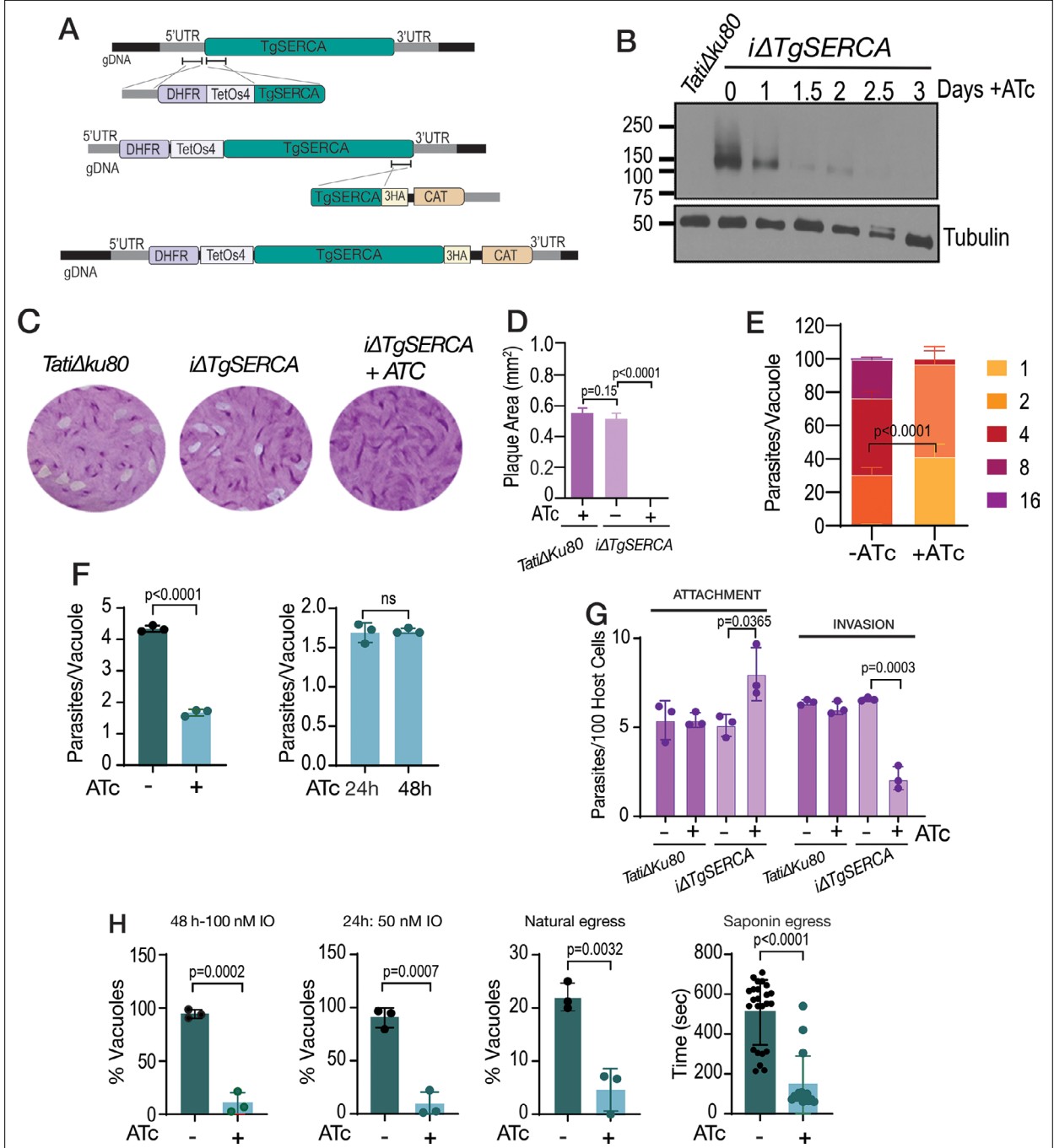

**Figure 3.** The sarco/endoplasmic reticulum Ca²⁺-ATPase (SERCA) is essential for the *T. gondii* lytic cycle. (**A**) Scheme showing the strategy used for generating conditional knockouts of TgSERCA by promoter insertion and regulation by 0.5 µg/ml Anhydrotetracyclin (ATc). The resulting mutants were named *iΔTgSERCA* or *iΔTgSERCA-3HA* (C-terminally HA-tagged). DHFR, dihydrofolate reductase gene (pyrimethamine selection); CAT, chloramphenicol acetyltransferase gene (chloramphenicol selection). (**B**) Western blots of *iΔTgSERCA-3HA* parasites grown ± ATc. TgSERCA expression was detected using an anti-HA antibody, showing reduced levels with ATc treatment. (**C**) Plaque assays comparing the growth of *iΔTgSERCA* tachyzoites (150 parasites/well) cultured ± 0.5 µg/ml ATc for 8 days. Plaques formed by the parental *TatiΔku80* strain are shown for comparison. (**D**) Quantification of the size of the plaques presented in C. (**E**) Replication assay using the *iΔTgSERCA-RFP* mutant. The number of parasites per parasitophorous vacuole (PV) was quantified 24 hr post-infection of fibroblast cells and compared between parasites grown ± 0.5 µg/ml ATc. (**F**) Average number of parasites per PV counted at 24 hr after the initial infection. The graph to the right shows the number of parasites per PV of the *iΔTgSERCA* (+ATc) for 24 or 48 hr after the initial infection. (**G**) Invasion assay of the *iΔTgSERCA* mutant following 24 hr of ATc treatment, performed using the red-green assay described in the Methods section. (**H**) Egress assays with fibroblast monolayers infected with *iΔTgSERCA-RFP* parasites for 24 or 48 hr. Egress was triggered with ionomycin (IO; 100 nM or 50 nM) or saponin (0.01%). Natural egress was monitored following treatment with 1 µM compound 1 as described in

*Figure 3 continued on next page*

*Figure 3 continued*

the Methods section. % Vacuoles: 100 X Number of vacuoles egressed/total vacuoles. Data (**D, E, F, G, H**) are presented as mean from at least three biological replicates ± SD. Statistical significance was assessed using an unpaired two-tailed t-test.

The online version of this article includes the following source data and figure supplement(s) for figure 3:

**Source data 1.** Source data for *Figure 3* data of growth, replication, invasion, and egress of the mutant *T. gondii* compared to control.

**Source data 2.** Original files for western blot analysis displayed in *Figure 3B*.

**Source data 3.** PDF file containing original western blots for *Figure 3B*, indicating the relevant bands and treatments.

**Figure supplement 1.** Regulation of the expression of *T. gondii* sarco/endoplasmic reticulum Ca²⁺-ATPase (TgSERCA).

**Figure supplement 1—source data 1.** Original files for western blot analysis displayed in *Figure 3—figure supplement 1A*.

**Figure supplement 1—source data 2.** PDF file containing original western blots for *Figure 3B*, indicating the relevant bands and treatments.

following pre-incubation with 1 µM compound 1 (*Donald et al., 2002*), previously shown to synchronize parasite exit (*Vella et al., 2021*), were both markedly reduced in ATc-treated parasites, underscoring the critical role of ER Ca²⁺ stores in supporting both ionophore-induced and spontaneous egress (*Figure 3H*, *IO, and natural egress*). Interestingly, however, egress induced by saponin in the presence of extracellular Ca²⁺ was accelerated in ATc-treated parasites (*Figure 3H*, *saponin egress*). This enhancement may result from a more rapid rise in cytosolic Ca²⁺, reaching the egress threshold more quickly due to impaired SERCA activity combined with ongoing Ca²⁺ leak from the ER (*Vella et al., 2021*). The saponin concentration used selectively permeabilizes the host cell membrane, allowing extracellular Ca²⁺ to enter the parasite cytosol without compromising the integrity of the parasite plasma membrane. This is consistent with previous observations showing that tachyzoites remain motile and exhibit Ca²⁺ oscillations under similar conditions (*Borges-Pereira et al., 2015*). The resulting rise in cytosolic Ca²⁺ within the parasite stimulates motility and triggers egress. To further examine this phenomenon, we directly compared the timing of egress between untreated and ATc-treated *iΔTgSERCA* parasites under identical saponin exposure conditions (*Figure 3H*, *Saponin egress*).

In summary, our findings demonstrate that TgSERCA is essential for *T. gondii* replication, invasion, and natural egress. Interestingly, when host cells were selectively permeabilized, parasites with reduced TgSERCA expression displayed accelerated egress, likely due to altered calcium dynamics.

## Ca²⁺ uptake by the SERCA-ATPase is essential for filling acidic Ca²⁺ stores

Further characterization of the *iΔTgSERCA* mutant showed a diminished cytosolic Ca²⁺ response to TG (*Figure 4A*), which was also observed when TG was applied after extracellular Ca²⁺ had been added to fill the stores (*Figure 4B*). This was most likely due to reduced Ca²⁺ accumulation by the ER in the *iΔTgSERCA* (+ATc) mutant. Note that the change in Ca²⁺ in *Figure 4A* (*TatiΔku80* cells) is larger than in *Figure 1B* (RH strain), which we attribute to differences between the two cell lines (RH vs *TatiΔku80*).

Most importantly, MgATP-driven Ca²⁺ uptake by permeabilized cells measured using Mag-Fluo-4, showed no TgSERCA activity after 48 hr of culture with ATc (*Figure 4C*). This experiment validated the Mag-Fluo-4 method for assessing SERCA activity. At 24 hr post-culture with ATc, some residual SERCA activity was still detected (*Figure 4C*).

Interestingly, Ca²⁺ entry measured in Fura-2 loaded *iΔTgSERCA* parasites (±ATc) was not affected by the downregulation of TgSERCA (*Figure 4D*). This finding argues against the presence of an ER-based mechanism that regulates Ca²⁺ entry. Moreover, the cytosolic resting Ca²⁺ concentration remained unchanged in the *iΔTgSERCA* (+ATc) mutant (*Figure 4A, B and D–H*) highlighting a critical role of the plasma membrane Ca²⁺ pump in maintaining cytosolic Ca²⁺ homeostasis.

The response to Zaprinast was diminished but was still present (*Figure 4E*) indicating that Zaprinast induces Ca²⁺ release from the ER and from an additional compartment. When Zaprinast was added after Ca²⁺ replenishment, the response remained reduced in the mutant pre-incubated with ATc (*Figure 4F*). We next tested GPN, which primarily targets acidic stores, and observed a decreased response (*Figure 4G*). Adding GPN after replenishing the cells with Ca²⁺ resulted in an increased

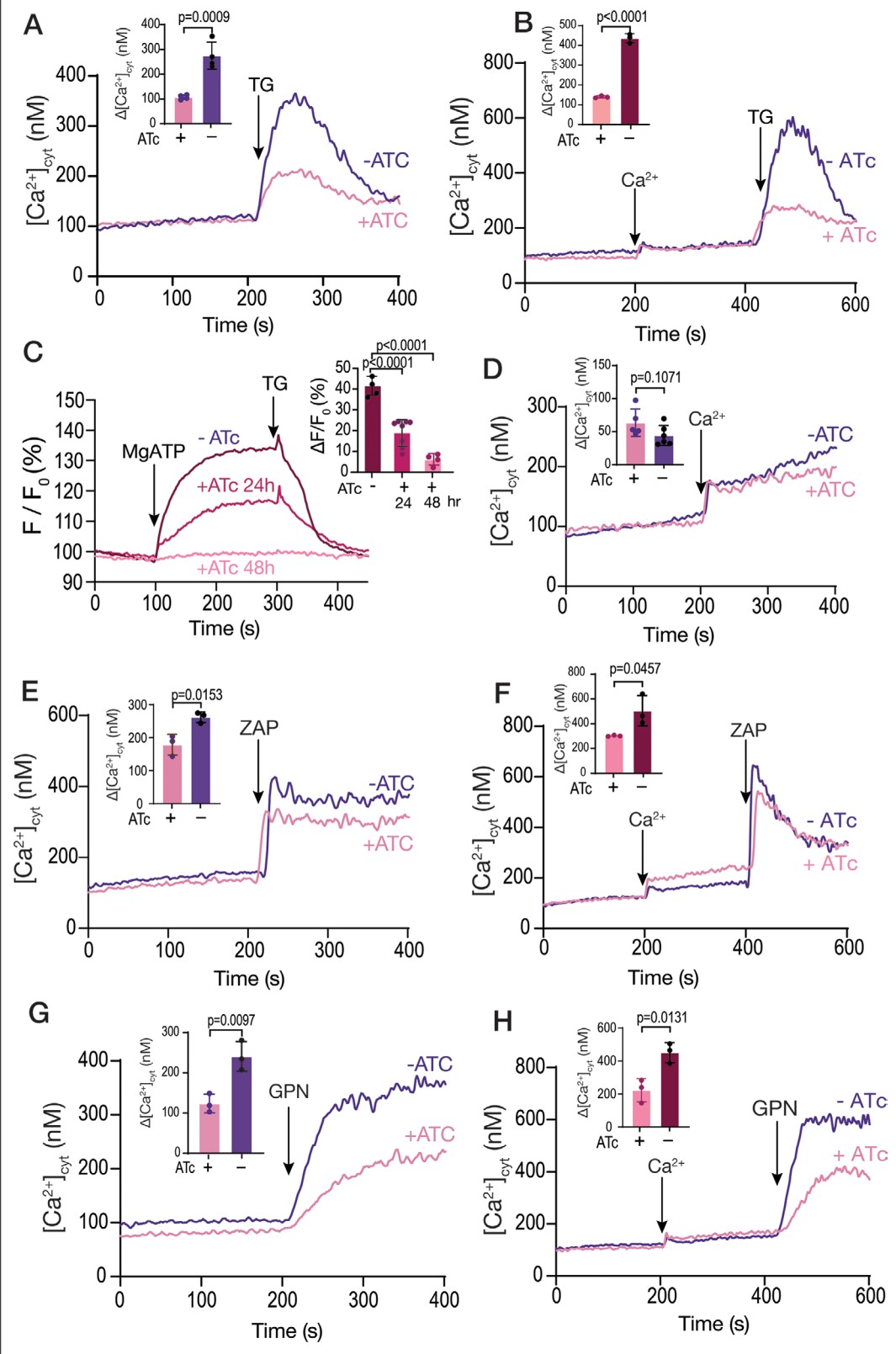

**Figure 4.** Organellar calcium pools in the *iΔTgSERCA* mutant. (**A**) The *iΔTgSERCA* mutant was grown ±ATc and was loaded with Fura-2 for cytosolic Ca²⁺ measurements. 1 μM thapsigargin (TG) was added at 200 s to a suspension of tachyzoites. The purple trace shows the response of the parental cell line grown without anhydrotetracycline (ATc) and the pink trace shows the response of the same mutant grown with ATc for 24 hr. The

*Figure 4 continued on next page*

*Figure 4 continued*

bar graph shows the analysis of the $\Delta[Ca^{2+}]_{cyt}$ from three biological experiments. (**B**) Same experimental setup as the one in A but adding 1.8 mM extracellular $Ca^{2+}$ at 200 s. (**C**) Sarco/endoplasmic reticulum $Ca^{2+}$-ATPase (SERCA) activity measured in Mag-Fluo-4 loaded *iΔTgSERCA* tachyzoites grown ±ATc. Parasites were collected, loaded with Mag-Fluo-4AM, and permeabilized with digitonin as described in the Methods section. Free $Ca^{2+}$ in the buffer was set at 220 nM, and MgATP (0.125 mM) was added at 100 s. The purple trace represents the control (no ATc), while the other traces correspond to parasites treated with ATc for 24 or 48 hr. TG (1 μM) was added as indicated. The bar graph shows the quantification of the initial slope after adding MgATP. (**D**) $Ca^{2+}$ entry measured in Fura-2– loaded *iΔTgSERCA* parasites grown ±ATc. Extracellular $Ca^{2+}$ (1.8 mM) was added at 200 s. The inset shows ΔF values from three independent experiments, indicating no significant differences. (**E**) Similar conditions to the ones used in A but adding 100 μM Zaprinast. The bar graph shows the quantification of the $\Delta[Ca^{2+}]$ from three biological experiments. (**F**) Similar conditions to the ones used in B but adding 1.8 mM extracellular calcium at 200 s and 100 μM Zaprinast at 400 s. The bar graph shows the quantification of the $\Delta[Ca^{2+}]$ from three biological experiments. (**G**) Same as A but adding 40 μM glycyl-L-phenylalanine-naphthylamide (GPN). The bar graph shows the analysis of the $\Delta[Ca^{2+}]$ from three biological replicates. (**H**) Same setup as in F but adding 1.8 mM $Ca^{2+}$ at 200 s followed by 40 μM GPN at 400 s. The bar graph shows the quantification of the $\Delta[Ca^{2+}]$ from three biological replicates. Data are presented as mean ±SD. *p*-value: unpaired two-tailed t-test performed in all comparisons.

The online version of this article includes the following source data for figure 4:

**Source data 1.** Source data for *Figure 4* showing calcium measurements with Fura2 and MagFluo4.

---

response, as we showed in *Figure 1*, but this response was also reduced when the mutant was grown with ATc (*Figure 4H*).

Given that the $Ca^{2+}$ phenotypes were assessed after 24 hr of ATc treatment, when approximately 50% of TgSERCA activity remains, the response to Zaprinast may still reflect ER involvement, and may not provide definitive evidence for the contribution of an additional $Ca^{2+}$ pool. To further investigate this, we conducted an experiment in which TG was added prior to GPN and Zaprinast. In this setting, GPN significantly reduced the Zaprinast-induced response (*Figure 1—figure supplement 2B*). This result suggests that Zaprinast also targets a non-ER $Ca^{2+}$ store, and that this store is likely the same one affected by GPN.

These results support a functional link between the stores targeted by GPN and the ER. Given that SERCA downregulation impaired ER $Ca^{2+}$ storage without affecting cytosolic $Ca^{2+}$ uptake and cytosolic $Ca^{2+}$ levels, the diminished response to GPN suggests that $Ca^{2+}$ released or leaked from the ER is important for refilling the store targeted by GPN.

## The mitochondrion takes up $Ca^{2+}$ from the ER and from acidic stores

In mammalian cells, the high concentration of $Ca^{2+}$ in the ER is important for mitochondrial ATP production (*Wenzel et al., 2022*). This is because of the close proximity between the ER and mitochondria which allows for the directional flow of $Ca^{2+}$ from the ER to the mitochondria (*Gincel et al., 2001*; *Rapizzi et al., 2002*). With the aim of verifying if the *T. gondii* mitochondria can take up calcium, we introduced a genetic $Ca^{2+}$ indicator in the mitochondrion of *T. gondii* tachyzoites by attaching the *GCaMP6f* gene (*Chen et al., 2013*) to the mitochondrial targeting signal of the *T. gondii* superoxide dismutase 2 (SOD2) gene (*Pino et al., 2007*) and isolated stable transgenic clones (RH-*SOD2-GCaMP6f*) (*Vella et al., 2020*). Fluorescence microscopy of live cells confirmed GCaMP6f localization to the mitochondria (*Figure 5A*). Direct $Ca^{2+}$ uptake was observed in digitonin-permeabilized parasites incubated in the presence of increasing concentrations of $Ca^{2+}$ (*Figure 5B–C*). Although a measurable increase in mitochondrial fluorescence was observed, it required high $Ca^{2+}$ concentrations, indicating that the *T. gondii* mitochondrion can take up $Ca^{2+}$ but do so with very low affinity. These $Ca^{2+}$ levels were significantly higher than the typical cytosolic $Ca^{2+}$ concentrations found in healthy cells (*Figure 5B–C*).

We hypothesized that the *T. gondii* mitochondrion may take up $Ca^{2+}$ through close membrane contacts with the ER, where localized $Ca^{2+}$ concentrations in microdomains could be significantly higher than in the cytosol, a mechanism previously described in mammalian cells (*Rizzuto et al., 1998*). We next loaded the RH-*SOD2-GCaMP6f* mutant with Fura-2 to simultaneously monitor cytosolic and mitochondrial $Ca^{2+}$ in intact parasites. Upon addition of extracellular $Ca^{2+}$, an increase in cytosolic $Ca^{2+}$ was observed, however, mitochondrial GCaMP6f fluorescence remained unchanged (*Figure 5D–E*), suggesting that mitochondria are unable to take up $Ca^{2+}$ at the cytosolic concentrations

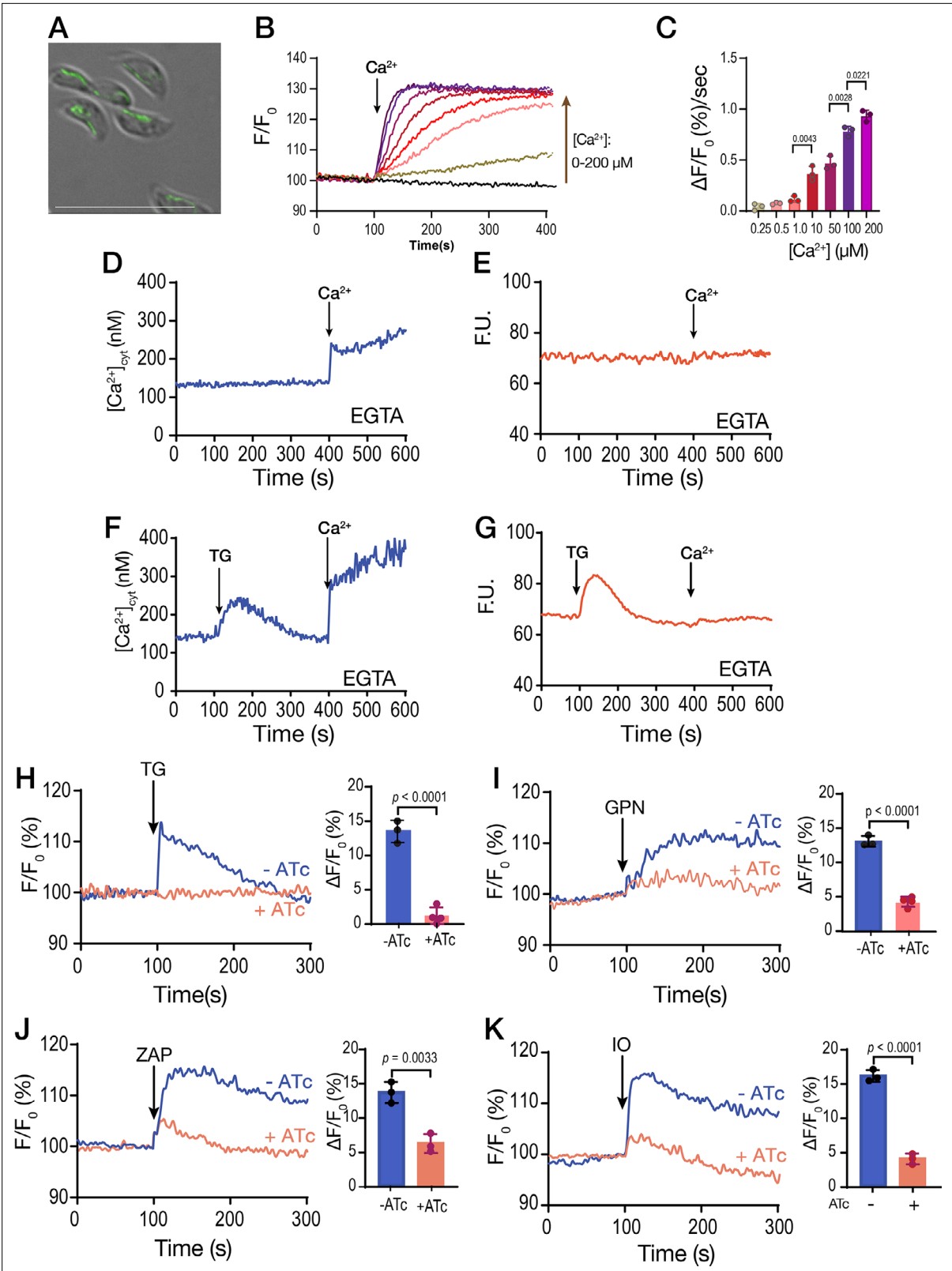

**Figure 5.** Mitochondrial calcium uptake. (**A**) Fluorescence image of *T. gondii* tachyzoites of the RH strain expressing *SOD2-GCaMP6f* (pDT7S4H3-SOD2-GCaMP6f). The generation of this cell line is described in the Methods section. Scale bar is 10 μm. (**B**) Ca²⁺ uptake in digitonin-permeabilized *T. gondii* tachyzoites expressing SOD2-GCaMP6f. Parasites (5×10⁷) were permeabilized as described in the Methods section and suspended in buffer containing 100 μM EGTA. Ca²⁺ was added at 100 s to reach final free concentrations of 0.25, 0.5, 1, 10, 50, 100, and 200 μM, calculated using

*Figure 5 continued on next page*

*Figure 5 continued*

Maxchelator. (**C**), ΔF was measured as the change in fluorescence between the baseline and the maximum value obtained 20 s after Ca²⁺ addition. Data represent the average of three independent biological experiments. (**D**) Fura-2-loaded *T. gondii* tachyzoites expressing SOD2-GCaMP6f in suspension. The experimental setup was identical to that described in *Figure 1A–B*. CaCl₂ (1.8 mM) was added at 400 s, and fluorescence measurements were performed under Fura-2 conditions. (**E**) GCaMP6f fluorescence measurements of the same parasites from D but the fluorescence was recorded using optimized settings for GCaMP6 detection. (**F**) Tachyzoites expressing SOD2-GCaMP6f loaded with Fura-2 in suspension. 1 µM thapsigargin (TG) was added at 100 s followed by 1.8 mM CaCl₂ at 400 s. Fura-2 conditions were used. (**G**) Same additions and same parasites as in F but measuring fluorescence of GCaMP6f. (**H**) Response to 1 µM TG of *iΔTgSERCA*-SOD2-GCaMP6f parasites (transfected with the pCTH3-SOD2-GCaMP6f plasmid), grown with (pink trace) or without (blue trace) anhydrotetracycline (ATc). Fluorescence measurements were performed under the same conditions as in panel G using intact parasites. The bar graph shows ΔF values from three independent biological replicates. (**I**) Same as H but using 40 µM glycyl-L-phenylalanine-naphthylamide (GPN). (**J**) Same as H but using 100 µM Zaprinast. (**K**) Same as H but using 1 µM Ionomycin (IO). Data are presented as mean ± SD from three independent biological experiments. *p-value*: unpaired two-tailed t-test performed in all comparisons.

The online version of this article includes the following source data and figure supplement(s) for figure 5:

**Source data 1.** Source data for *Figure 5* showing Fura 2 and GCaMP6 calcium measurements.

**Figure supplement 1.** Mitochondrial localization of the GCaMP6f.

reached under these conditions. This also validates the proper localization of the indicator, confirming its absence from the cytosol. Addition of TG followed by extracellular Ca²⁺ resulted in a cytosolic Ca²⁺ increase, readily detected in Fura-2-loaded parasites. However, and most importantly, only TG triggered a measurable increase in the mitochondrial GCaMP6f signal, whereas a rise in cytosolic Ca²⁺ induced by extracellular Ca²⁺ addition alone did not (*Figure 5G*). Our interpretation is that TG-induced ER Ca²⁺ leakage led to local accumulation of Ca²⁺ at the cytosolic face of the ER membrane, creating microdomains of high Ca²⁺ concentration sufficient to trigger mitochondrial uptake. Addition of Ca²⁺ after TG resulted in a greater increase in cytosolic Ca²⁺ (*Figure 5F*) compared to TG alone. However, even under these conditions, no corresponding increase in mitochondrial GCaMP6f fluorescence was observed. This further confirms that mitochondria are unable to take up cytosolic Ca²⁺ at these low concentrations.

We next introduced the same *SOD2-GCaMP6f* chimeric gene into the *iΔTgSERCA* mutant background and isolated a clonal line (*iΔTgSERCA-SOD2-GCaMP6f*) (*Figure 5—figure supplement 1A–B*). Fluorescence imaging of live cells confirmed proper localization of the indicator (*Figure 5—figure supplement 1A*), and fluorescence measurements of intact cells corroborated that adding extracellular Ca²⁺ did not increase GCaMP6f fluorescence, whereas addition of TG to the suspension resulted in a fluorescence increase (*Figure 5—figure supplement 1B*). We next monitored changes in GCaMP6f fluorescence in the mutant and compared results between parasites grown with and without ATc. In line with prior observations, parasites cultured without ATc showed a consistent and measurable increase in mitochondrial GCaMP6f signal upon TG treatment (*Figure 5H*, *blue trace*). In contrast, this response was abolished in parasites cultured with ATc, consistent with reduced TgSERCA expression leading to ER Ca²⁺ depletion (*Figure 5H*, *pink trace*).

We next tested additional stimuli and observed a clear increase in mitochondrial GCaMP6f fluorescence in the *iΔTgSERCA* mutant following the addition of GPN, Zaprinast, or IO (*Figure 5I-K*, *blue traces*). In all cases, this fluorescence increase was significantly reduced in the *iΔTgSERCA* (+ATc) mutant (*Figure 5I-K*, *pink traces*). These results suggest a potential direct interaction between the mitochondrion and acidic Ca²⁺ stores, such as the PLVAC and/or Golgi apparatus. The reduced Ca²⁺ content of these compartments, resulting from TgSERCA downregulation, appears to impact mitochondrial Ca²⁺ uptake.

In summary, we demonstrated that the *T. gondii* mitochondrion is capable of Ca²⁺ uptake via transfer from the ER, a process that becomes apparent upon inhibition of ER Ca²⁺ uptake with TG. This suggests that the high Ca²⁺ concentrations required for mitochondrial uptake are achieved only at membrane contact sites between the ER and mitochondria. In the *iΔTgSERCA* (+ATc) mutant, impaired ER Ca²⁺ storage due to TgSERCA downregulation compromises mitochondrial Ca²⁺ uptake. Additionally, our data suggest that the *T. gondii* mitochondrion may also take up Ca²⁺ from acidic stores, such as the PLVAC or Golgi, which appear to rely indirectly on ER Ca²⁺ refilling. When TgSERCA is downregulated, depletion of ER Ca²⁺ likely compromises the Ca²⁺ content of these acidic compartments, and impairing mitochondrial Ca²⁺ uptake from these stores.

## Proximity between the ER, mitochondrion, and acidic compartment

We next investigated whether proximity between the ER and other organelles could be detected by IFA and/or electron microscopy (EM). We performed IFAs with ER and mitochondria markers and ER and PLVAC markers (*Figure 6*). In intracellular parasites, the mitochondrion was observed to surround the ER, forming multiple potential sites of interaction (*Figure 6A* and *Figure 6—video 1*). As previously described, the mitochondrion of intracellular *T. gondii* tachyzoites surrounds the periphery of the cell in a lasso-shape morphology (*Ovciarikova et al., 2017*). In contrast, in extracellular parasites, the mitochondrion displays a marked morphological change, adopting either a sperm-like or collapsed conformation (*Ovciarikova et al., 2017*; *Figure 6B* and *Figure 6—video 2*). Our hypothesis is that retraction of the mitochondrion allows the ER membranes to expand in extracellular parasites and extend toward the apical domain, where $Ca^{2+}$ is required for micronemes secretion and conoid extrusion.

The PLVAC also formed points of contact with the ER (*Figure 6C* and *Figure 6—video 3*). Multiple points of contact were also observed by EM between the ER and the PLVAC, the ER and the apicoplast, and the ER and the mitochondrion (*Figure 6D*).

Interestingly, these contacts were still present in the *iΔTgSERCA* (+ATc) mutant (*Figure 6—figure supplement 1A–D*), as most likely TgSERCA would not be directly involved in the establishment of contacts. We quantified the length of the limiting membrane of the organelle in contact with ER membranes at a distance of less than 30 nm and found that after knockdown of TgSERCA, the contact was not altered. However, $Ca^{2+}$ transported from the ER into the mitochondrion after TG treatment was significantly decreased which means that this phenotype is due to reduced ER $Ca^{2+}$ and not to lack of contacts (*Figure 6—figure supplement 1E*). A similar result was seen when measuring contacts between the ER and the apicoplast (*Figure 6—figure supplement 1F*) and between the ER and the PLVAC (*Figure 6—figure supplement 1G*).

In summary, this data supports the presence of points of contact between the ER and other organelles like the mitochondrion, the PLVAC, and the apicoplast. These contacts likely facilitate the transfer of $Ca^{2+}$ from the ER, the organelle with the highest $Ca^{2+}$ content, to other compartments.

## Discussion

In this work, we demonstrated that the ER of *T. gondii* has a remarkable capacity to sequester $Ca^{2+}$ entering the cytosol from the extracellular milieu, achieving this with only a minimal rise in cytosolic $Ca^{2+}$ levels. This is largely due to the activity of a highly efficient SERCA $Ca^{2+}$-ATPase (TgSERCA), which has a high affinity for $Ca^{2+}$. The activity of TgSERCA, most likely together with the plasma membrane $Ca^{2+}$ pump (*Luo et al., 2001*; *Luo et al., 2005*), limits large increases in cytosolic $Ca^{2+}$ (*Hortua Triana et al., 2024*).

We provide evidence that the ER not only sequesters extracellular $Ca^{2+}$ through TgSERCA activity but also shares this pool with other organelles, including mitochondria and acidic stores. This capacity stems from the unique ability of the ER to capture a sizable fraction of extracellular $Ca^{2+}$ entering the tachyzoite cytosol. Such inter-organelle transfer allows localized $Ca^{2+}$ release without globally elevating cytosolic levels, thereby preventing unintended signaling events. Our data support a model in which loss of SERCA activity reduces ER $Ca^{2+}$ as well as $Ca^{2+}$ content in other organelles. Under physiological conditions, ER $Ca^{2+}$ is regularly mobilized for signaling and homeostasis, helping to maintain $Ca^{2+}$ balance across cellular compartments (see our hypothetical model in *Figure 7*).

SERCA $Ca^{2+}$-ATPases are P-type pumps located in the ER and secretory pathway membranes (*Kühlbrandt, 2004*). Mammals express three isoforms (SERCA1–3), with SERCA2b serving as the housekeeping form (*Wuytack et al., 2002*). SERCA pumps translocate two $Ca^{2+}$ ions into the ER lumen per ATP hydrolyzed, lowering cytosolic $Ca^{2+}$ to resting levels (<100 nM) and replenishing ER stores (~500 µM). This stored $Ca^{2+}$ supports signaling and the activity of luminal enzymes critical for cell growth, proliferation, and differentiation (*Wuytack et al., 2002*).

*T. gondii* appears to express a single SERCA protein (TgSERCA) (*Nagamune and Sibley, 2006*), likely serving a housekeeping role. The severe defects observed in the *iΔTgSERCA* (+ATc) mutant like impaired replication and disruption of the lytic cycle, highlight its essential function. TgSERCA activity was dependent on MgATP and exhibited high $Ca^{2+}$ affinity, as evidenced by Mag-Fluo-4-based assays detecting uptake at free $Ca^{2+}$ levels as low as 55 nM. This suggests that TgSERCA functions effectively

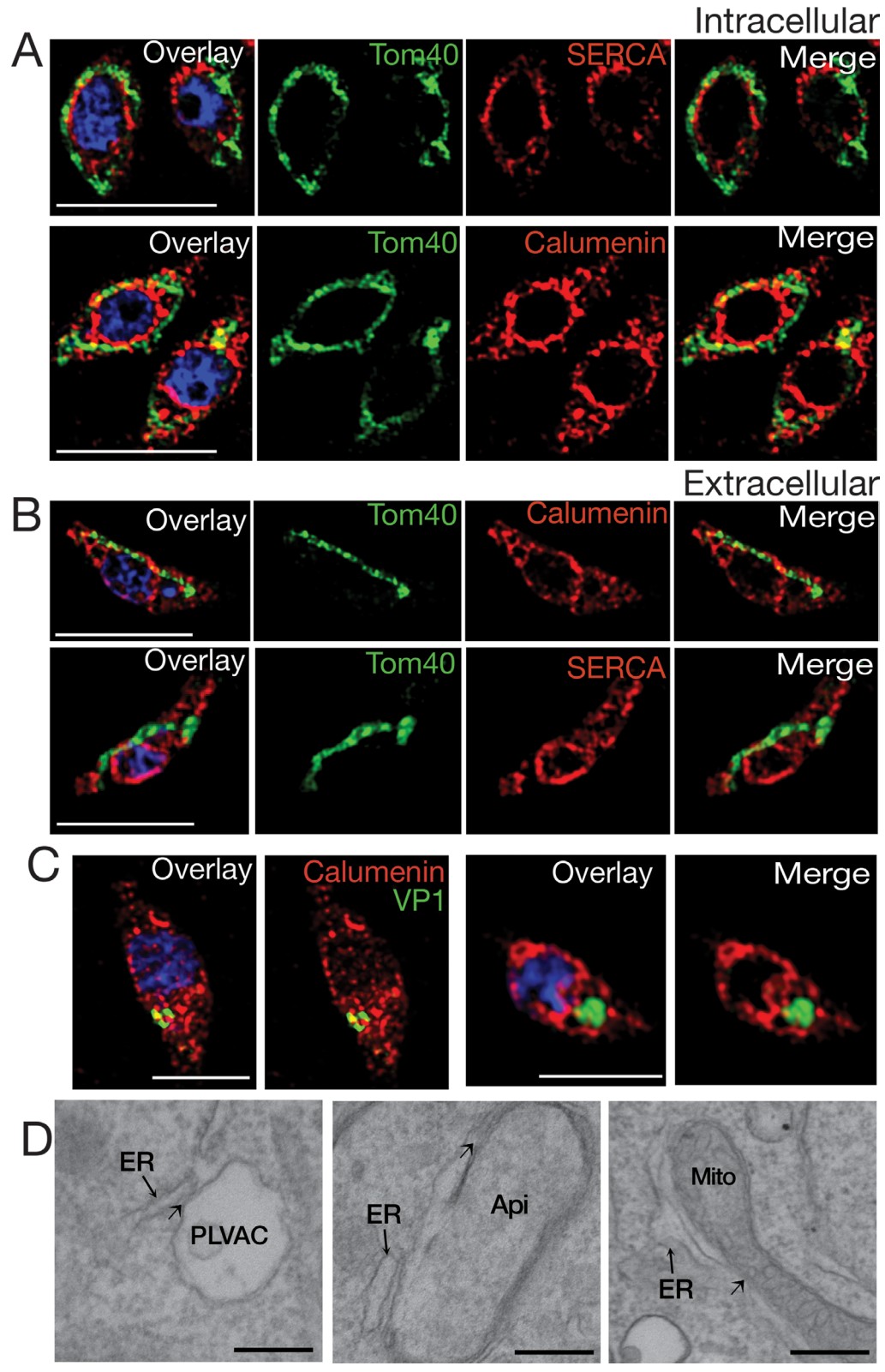

**Figure 6.** Endoplasmic reticulum (ER)-mitochondria-Plant-Like Vacuolar Compartment (PLVAC) associations revealed by immunofluorescence and electron microscopy. (**A**) Super-resolution IFAs of intracellular parasites with the mitochondrion labeled with the αTom40 (green, 1:20,000) antibody and the ER labeled with the αTgcalumenin antibody (an ER calcium binding protein) (red, 1:1,000) or the αTgSERCA (red 1:1,000). (**B**) IFAs of extracellular

*Figure 6 continued on next page*

*Figure 6 continued*

tachyzoites with the same antibodies used for part A. Close associations between the mitochondrial and ER membranes are observed at several regions. (**C**) The PLVAC was labeled with the αTgVP1 antibody (green, 1:200) or the αTgCPL antibody (green, 1:500). The ER was labeled with the αTgcalumenin antibody (red). The points of contact between the ER and the PLVAC are yellow. Scale bars in A-C are 5 μm. (**D**) Transmission Electron Microscopy imaging of the contact sites formed between ER and PLVAC, ER and Apicoplast, ER and mitochondria. Scale bars are 100 nm.

The online version of this article includes the following video and figure supplement(s) for figure 6:

**Figure supplement 1.** Electron microscopy images of the *iΔTgSERCA* mutant treated with anhydrotetracycline (ATc) for 24 hr.

**Figure 6—video 1.** IFA of intracellular parasites labeled with the mitochondria marker αTom40 (green) antibody, and the ER labeled with the αTgERC antibody (red).
https://elifesciences.org/articles/101894/figures#fig6video1

**Figure 6—video 2.** Imaris 3D optimal visualization of extracellular parasites labeled with the αTom40 (green) antibody and the endoplasmic reticulum (ER) labeled with the αTgERC antibody (red).
https://elifesciences.org/articles/101894/figures#fig6video2

**Figure 6—video 3.** Endoplasmic reticulum (ER) membrane contacts sites with the Plant-Like Vacuolar Compartment (PLVAC).
https://elifesciences.org/articles/101894/figures#fig6video3

at physiological cytosolic $Ca^{2+}$ concentrations (60–100 nM), ensuring ER loading even under resting conditions.

*In situ* characterization of organellar $Ca^{2+}$ uptake has been feasible in trypanosomes (*Docampo and Vercesi, 1989*; *Vercesi et al., 1990*) but remains challenging in *T. gondii*. The Mag-Fluo-4 and the mitochondrial GCaMP6f protocols enable reliable measurement of ER and mitochondrial $Ca^{2+}$ uptake, respectively. For ER $Ca^{2+}$ uptake, MgATP was essential for the activity of TgSERCA, as other forms of ATP were ineffective. This protocol has been extensively used in mammalian cells, DT40 cells and other cells for measuring intraluminal calcium, activity of SERCA, and response to $IP_3$ (*Laude et al., 2005*; *Rossi et al., 2009*; *Valverde et al., 2010*; *Sampieri et al., 2018*; *Rossi and Taylor, 2020*). We previously successfully employed it for the characterization of the *Trypanosoma brucei* $IP_3R$ (*Huang et al., 2013*) and the assessment of SERCA activity in *T. gondii* mutants (*Li et al., 2021*). In this work, we used it to assess TgSERCA activity under defined $Ca^{2+}$ and MgATP conditions.

Using the Mag-Fluo-4 protocol, we observed that cyclopiazonic acid (CPA), a reversible SERCA inhibitor (*Inesi and Sagara, 1994*), induced a $Ca^{2+}$ leak rate comparable to the one after adding TG. This indicates that the leak rate is mainly determined by intrinsic leak mechanisms rather than the type of SERCA inhibition. However, in intact Fura-2-loaded parasites, CPA induced a smaller cytosolic $Ca^{2+}$ increase than TG. This likely reflects CPA's reversible and potentially incomplete inhibition of SERCA under cellular conditions, as was also observed in *Plasmodium falciparum* (*Borges-Pereira et al., 2020*).

We observed that the response to acidic calcium triggers like nigericin or GPN were greatly enhanced when added after TG in Fura-2-loaded tachyzoites, likely due to ER $Ca^{2+}$ leak and subsequent transfer to other compartments. Additionally, downregulation of *TgSERCA* expression resulted in reduced responses to these acidic store triggers, supporting the notion that the ER contributes to the filling of these organelles. It is important to note that our analyses of $Ca^{2+}$ storage was done in parasites that retained partial TgSERCA activity, as it is not possible to isolate cells entirely lacking TgSERCA expression. Under these conditions, Zaprinast still induced a reduced $Ca^{2+}$ mobilization response. This residual response may be due to remaining calcium in the ER or may suggest that Zaprinast targets multiple calcium stores. We recently identified the Golgi apparatus as a calcium store in *T. gondii* (*Calixto et al., 2025*) and demonstrated that treatment with GPN in Fura-2-loaded tachyzoites diminished the Zaprinast-induced calcium response, suggesting that Zaprinast and GPN may act on overlapping stores. In the present study, we demonstrated that sequential treatment with TG followed by GPN almost completely abolished the Zaprinast response, further supporting this idea. Although GPN is primarily known to act on acidic organelles, it has also been proposed to affect the ER (*Atakpa et al., 2019*) however, we have no evidence that GPN mobilizes calcium from the ER

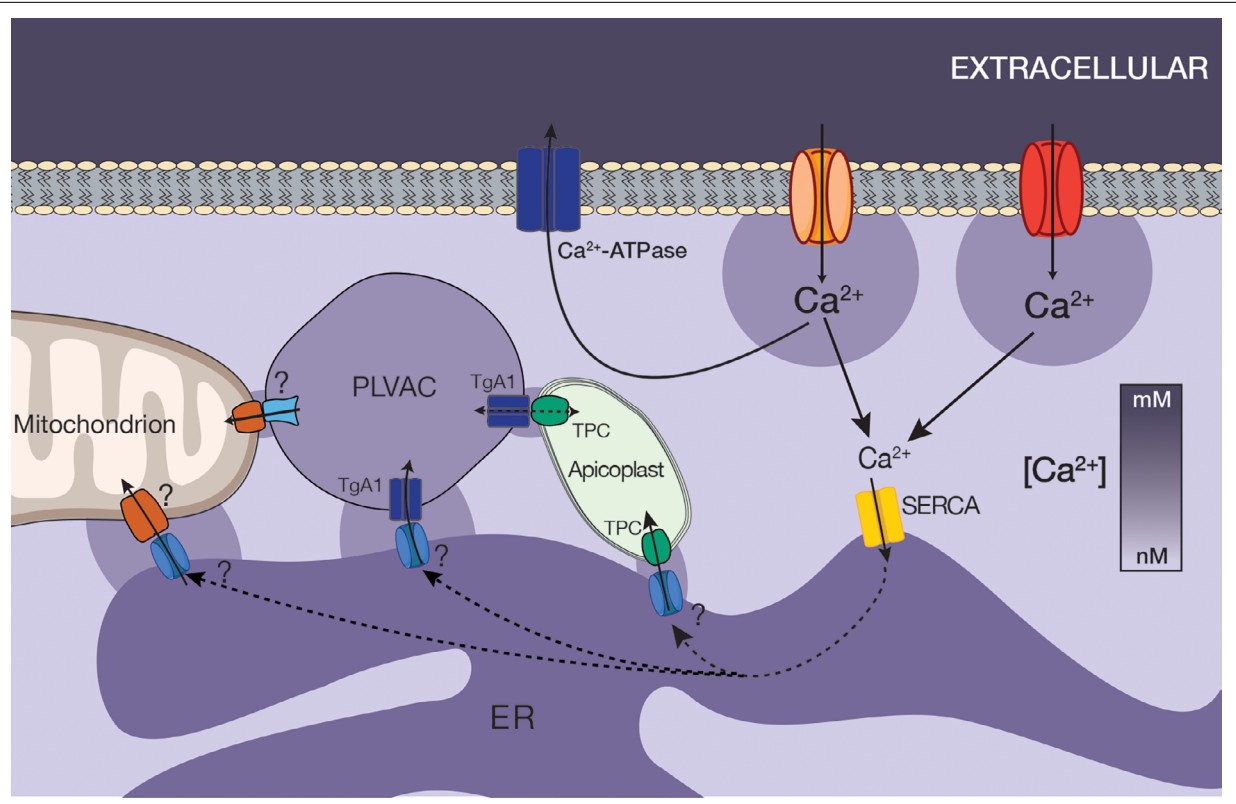

**Figure 7.** Hypothetical model showing Ca²⁺ entry through two different types of Ca²⁺ channels, uptake by *T. gondii* sarco/endoplasmic reticulum Ca²⁺-ATPase (TgSERCA) into the endoplasmic reticulum (ER) and distribution to the other organelles via transfer from the ER to the mitochondria, Plant-Like Vacuolar Compartment (PLVAC), and apicoplast. The mitochondrion is shown in close contact to the ER which constitutively leaks Ca²⁺ into the cytosol. Ca²⁺ could leak from the ER through the TgTRPPL-2 channel previously described (*Márquez-Nogueras et al., 2021*). The mitochondria take up Ca²⁺ from the ER through an unknown mechanism. Voltage-dependent anion channel (VDAC) could be involved in the transfer through the outer mitochondrial membrane (*Mallo et al., 2021*). The PLVAC interacts with the ER and may also interact with the mitochondrion and the apicoplast. TgA1, a calcium ATPase previously characterized may be the pump involved in Ca²⁺ uptake (*Luo et al., 2001*; *Luo et al., 2005*). The mechanism of release is unknown. The Two Pore Channel (TgTPC) was shown to be involved in the mechanism of transfer of Ca²⁺ between the ER and the apicoplast (*Li et al., 2021*). Question marks point to molecules or mechanisms partially or not yet identified.

in *T. gondii*. We propose that GPN primarily targets the PLVAC, but further investigation is required to fully characterize its mechanism of action.

It was interesting that Ca²⁺ entry remained unchanged in the *iΔTgSERCA* (+ATc) mutant, suggesting that intracellular stores may not be directly involved in the regulation of Ca²⁺ entry (*Pace et al., 2014*). Moreover, earlier genomic analysis did not identify clear homologs of the canonical Store-Operated Calcium Entry (SOCE) components STIM and Orai (*Collins and Meyer, 2011*), raising the possibility that these proteins are either absent or highly divergent in sequence and lack conserved regulatory domains. If communication between intracellular stores and the plasma membrane exists in *T. gondii*, the underlying mechanism remains unclear.

Calcium transport into the *T. gondii* mitochondrion had not been previously demonstrated and our findings provide the first experimental evidence for this process, though the molecular mechanism remains unclear. We found that normal cytosolic Ca²⁺ fluctuations were insufficient to drive mitochondrial uptake, consistent with the low Ca²⁺ affinity of the mitochondrion. Uptake occurred only after SERCA inhibition, which caused local Ca²⁺ accumulation at the cytosolic side of the ER membrane, enabling transfer to the mitochondrion, likely via membrane contact sites (MCSs), since direct uptake from the cytosol would be inefficient at low Ca²⁺ concentrations. MCSs were defined as stable, tightly apposed, but non-fusogenic regions of close proximity between subcellular organelles, and play a key role in inter-organelle communication (*Phillips and Voeltz, 2016*). In mammalian cells, the ER forms an extensive network of MCSs with the PM, mitochondria, and endocytic vesicles for the exchange of Ca²⁺ (*Burgoyne et al., 2015*).

In *T. gondii,* the characterization of MCSs is only in its beginnings (*Huet and Moreno, 2023*) with only a few evidences for their presence (*Tomova et al., 2009*; *Mallo et al., 2021*; *Ovciarikova et al., 2022*; *Ovciarikova et al., 2024*) and function (*Li et al., 2021*; *Oliveira Souza et al., 2022*). Imaging of intracellular *T. gondii* showed that its mitochondrion surrounds the periphery of the cell in a lasso-shape conformation. On the other hand, in extracellular parasites, the mitochondrion changes its morphology and adopts a sperm-like or collapsed conformation (*Ovciarikova et al., 2017*). Our IFA analysis with ER and mitochondrial markers revealed that the lasso-shaped mitochondrion surrounds the ER with plenty of opportunities for contact between both organelles.

Interestingly, the mitochondrion also appeared capable of importing $Ca^{2+}$ from acidic stores such as the PLVAC, as GPN treatment stimulated mitochondrial $Ca^{2+}$ uptake. This response was reduced in the *iΔTgSERCA* (+ATc) mutant, indicating that TgSERCA activity contributes to the transfer of $Ca^{2+}$ from acidic stores to the mitochondrion. These findings suggest a functional interdependence among intracellular $Ca^{2+}$ stores and highlight a central role for the ER in coordinating $Ca^{2+}$ dynamics.

In mammalian cells, $Ca^{2+}$ ion is transferred from the ER to the mitochondrion through the outer membrane voltage-dependent anion channel1 (VDAC1) (*Gincel et al., 2001*; *Rapizzi et al., 2002*) and the inner membrane calcium uniporter (MCU1) (*De Stefani et al., 2011*). A VDAC homologue is present in *T. gondii*, which was shown to be essential for growth and for mitochondrial and ER morphology (*Mallo et al., 2021*). However, molecular evidence for the presence of a $Ca^{2+}$ uniporter in the inner mitochondrial membrane, driven by the electrochemical gradient generated by the electron transport chain, remains to be demonstrated.

Cellular responses triggered by $Ca^{2+}$ signals are shaped by the location, duration, and amplitude of the signals. Movement of $Ca^{2+}$ in the cytosol of cells is severely limited due to the presence of high-affinity $Ca^{2+}$ buffers. In mammalian cells, it was shown that $Ca^{2+}$ tunnels through the ER as it moves faster because the $Ca^{2+}$ binding capacity of the ER is almost 100 times lower than the binding capacity of the cytosol (*Mogami et al., 1997*). The ER $Ca^{2+}$ transport through its lumen was shown to provide a mechanism for delivering $Ca^{2+}$ to targeted sites without activating inappropriate processes in the cell cytosol (*Petersen et al., 2017*). In *T. gondii*, the relative $Ca^{2+}$-binding capacity of the cytosol compared to the ER remains poorly understood, as the localization of many predicted $Ca^{2+}$-binding proteins has not been fully determined. Several calmodulin-like proteins, for example, are localized to the conoid (*Long et al., 2017*).

The mechanisms of $Ca^{2+}$ entry at the plasma membrane, release from the ER, and uptake by the mitochondria or acidic stores remain incompletely characterized (*Hortua Triana et al., 2018*; *Pace et al., 2020*; *Garcia et al., 2017*). Consequently, the molecular elements required for classical $Ca^{2+}$ tunneling have not been identified in *T. gondii*. Nevertheless, our results demonstrate that $Ca^{2+}$ can be transferred from the ER to other organelles. This is supported by the increased mitochondrial and acidic calcium pools observed following pharmacological ER depletion, both in the presence and absence of extracellular calcium. Importantly, chronic ER calcium depletion, such as in the *iΔTgSERCA* mutant cultured with ATc, leads to the depletion of all intracellular $Ca^{2+}$ stores. Additionally, we directly demonstrated mitochondrial $Ca^{2+}$ uptake when $Ca^{2+}$ accumulated on the cytosolic side of the ER membrane following SERCA inhibition. The specific roles of $Ca^{2+}$ in the mitochondrion and acidic compartments remain unclear. In mitochondria, $Ca^{2+}$ may support ATP production, although this has yet to be confirmed. Both organelles may also act as auxiliary $Ca^{2+}$ reservoirs during ER $Ca^{2+}$ overload.

In *T. gondii*, cytoplasmic $Ca^{2+}$ increases, due to efflux from the ER or entry through the PM, have been reported to initiate key parasite processes such as microneme secretion (*Carruthers and Sibley, 1999*; *Nagamune et al., 2007b*), conoid extrusion (*Del Carmen et al., 2009*; *Pace et al., 2014*), invasion (*Vieira and Moreno, 2000*; *Lovett and Sibley, 2003*), and egress (*Arrizabalaga et al., 2004*; *Borges-Pereira et al., 2015*). These responses require precise spatiotemporal regulation of $Ca^{2+}$ at specific cellular sites, suggesting the presence of mechanisms that direct $Ca^{2+}$ to discrete locations. We propose that the ER plays a central role in this regulation by acting as a hub that distributes $Ca^{2+}$ to defined sites at defined times to initiate parasite functions. The severe invasion, replication, and egress defects observed in the *iΔTgSERCA* (+ATc) mutant support this hypothesis.

Egress was one of the first steps of the *T. gondii* lytic cycle that was shown to be triggered by exposure of intracellular parasites to ionophores (*Endo et al., 1982*). Most recent work using GECIs demonstrated the rise in cytosolic calcium preceding egress (*Borges-Pereira et al., 2015*; *Stewart et al., 2017*). In the present study, we demonstrate that egress is defective and unresponsive to

ionophores in parasites lacking sufficient Ca²⁺ in their intracellular stores. This underscores the critical role of TgSERCA in maintaining Ca²⁺ stores filled. Interestingly, host cell permeabilization allowing extracellular Ca²⁺ entry rescued the defect, restoring and accelerating parasite egress.

In conclusion, this study demonstrates that the ER of *T. gondii* can replenish itself with Ca²⁺ and acts as a source of Ca²⁺ for cytosolic signaling, as well as for loading acidic stores and the mitochondrion. *T. gondii* is a protozoan parasite that causes disease by reiterating a lytic cycle that is driven by Ca²⁺ signaling. Our findings enhance understanding of how extracellular and intracellular Ca²⁺ stores coordinate to sustain the pathologic features of *T. gondii*. Future studies will focus on defining the roles of Ca²⁺ in mitochondrial and acidic stores functions.

## Methods

### Cell culture

*Toxoplasma gondii* tachyzoites (RH and *TatiΔku80* strain) were maintained in human telomerase reverse transcriptase immortalized foreskin fibroblasts (hTERT) (*Farwell et al., 2000*) grown in Dulbecco's modified minimal essential media (DMEM) with 1% FBS. These cells are tested for Mycoplasma contamination regularly and are treated with mycoplasma removal agent. The hTERT cell line (ATCC CRL-3627) is the only human cell line used in this project. It was obtained directly from ATCC and expanded in the absence of any other mammalian cells. After amplification, the cells were cryopreserved in liquid nitrogen. A single vial is thawed and used for approximately six months before being discarded and replaced with a new vial. The hTERT cells were used solely for the growth of *Toxoplasma gondii* tachyzoites. No experiments were performed using this cell line itself.

### Generation of SERCA mutants

A promoter insertion plasmid was generated by cloning three PCR fragments into a modified pCR2.1-TOPO vector using the Gibson Assembly Cloning Kit (NEB #E5510). One fragment corresponding to the TgSERCA flanking region (predicted promoter/5'UTR) was amplified with primers 1 and 2 (*Supplementary file 1*). The second fragment corresponds to DHFR + T7 S4 (*Sheiner et al., 2011*) and was amplified with primers 3 and 4. Another fragment corresponds to the 5' TgSERCA coding sequence beginning with start codon and was amplified with primers 5 and 6. The vector pCR2.1-TOPO was used, which had only one EcoRI site and it was cut with the enzyme NotI to use as vector backbone. The promoter insertion plasmid was transfected into the *TatiΔku80* cells and selected with 1 μM pyrimethamine using an 'ultra-aggressive' screening method. Briefly, 200 μl of the suspension of transfected parasites was added to 10 ml of medium, and one to three drops (~65 μl per drop) were inoculated into each well of three 24-well plates already filled with medium. The clonal lines created after selection and subcloning were termed *iΔTgSERCA*.

For *in situ* tagging, an approximately 2 kb fragment was amplified from the genomic locus (3' region) of the *TgSERCA* gene using primers 7 and 8. The fragment was cloned into the pLic-3HA-CAT plasmid (*Huynh and Carruthers, 2009*) and the construct was linearized with the enzyme NheI for transfection into the *iΔTgSERCA* mutant. Clonal cell lines were generated after selection with chloramphenicol and subcloning and termed *iΔTgSERCA-3HA*.

### Expression and purification of TgSERCA recombinant protein

The phosphorylation (P) and nucleotide binding (N) domains of *TgSERCA* (TGGT1_230420) (nucleotides 1123–2415, amino acid residues 375–805) were cloned into XmaI and HindIII sites of pQE-80L with primers 13 and 14 (*Supplementary file 1*) to create recombinant protein with a N-terminal 6xHis tag. The resulting plasmid was transformed into *Escherichia coli* BL21-CodonPlus competent cells and expression was induced by addition of 0.4 mM isopropyl β-D-1-thiogalactopyranoside (IPTG) for 4 hr at 37 °C. Cells were pelleted and resuspended in equilibration/binding buffer (50 mM Na₃PO₄, 300 mM NaCl, 10 mM Imidazole, 8 M Urea, and protease inhibitor cocktail, Sigma, P-8849). The cells were then sonicated for 80 s and centrifuged at 12,000 rpm for 20 min at 4 °C. The supernatant was filtered through a 0.45 μm membrane and the protein was purified using HisPur Ni-NTA Chromatography Cartridge (Thermo Scientific) following instructions from the manufacturer. Proteins that were unbound were washed with 12 ml of wash buffer (50 mM Na₃PO₄, 300 mM NaCl, 40 mM imidazole, and 8 M urea), and the recombinant protein was eluted with 5 ml elution buffer (50 mM Na₃PO₄,

300 mM NaCl, 250 mM imidazole, and 8 M urea). Eluted protein fractions were concentrated and desalted with an Amicon Ultra-0.5 mL centrifugal filter (Millipore Sigma).

## Anti-TgSERCA antibody generation in guinea pigs

Two guinea pigs were each immunized with 0.2 mg of purified recombinant protein mixed with equal volume of Freund's Complete Adjuvant (Sigma F5581), followed by two boosts of 0.1 mg antigen mixed with equal volume of Freund's Incomplete Adjuvant (Sigma F5506) for guinea pig 1 and three boosts for guinea pig 2. The resulting antibodies were tested at 1:1,000 in western blot against RH lysates and were developed with Alexa Fluor 488 goat anti-guinea pig (1:1,000). The antibodies were compared with Dr. Sibley's mouse anti-SERCA antibody (*Nagamune et al., 2007a*) to confirm size and purity (*Figure 3—figure supplement 1A*). The anti-TgSERCA antibodies were then affinity-purified. Guinea pigs were handled according to our approved institutional animal care and use committee (IACUC) protocols (A2021 03–005 A5) of the University of Georgia.

## Cytosolic calcium measurements with Fura-2

*T. gondii* tachyzoites were loaded with Fura-2 AM as previously described (*Vella et al., 2020*; *Stasic et al., 2021*). Freshly released tachyzoites were washed twice with buffer A plus glucose (BAG; 116 mM NaCl, 5.4 mM KCl, 0.8 mM MgSO4, 50 mM HEPES, pH 7.3, and 5.5 mM glucose), by centrifugation (706×$g$ for 10 min) and re-suspended to a final density of $1\times10^9$ parasites/ml in loading buffer (BAG plus 1.5% sucrose, and 5 µM Fura-2-AM). The suspension was incubated for 26 min at 26 °C with mild agitation. Subsequently, the parasites were washed twice (2,000×$g$ for 2 min) with BAG to remove extracellular dye, re-suspended to a final density of $1\times10^9$ parasites per ml in BAG and kept on ice. This loading protocol is specifically designed to minimize Fura-2 compartmentalization, which is typically indicated by elevated resting $Ca^{2+}$ concentrations. All experiments are conducted within a time frame during which resting $Ca^{2+}$ levels remain stable, typically below or at 100 nM. For fluorescence measurements, $2\times10^7$ parasites/mL were placed in a cuvette with 2.5 mL of Ringer's buffer without calcium (155 mM NaCl, 3 mM KCl, 1 mM $MgCl_2$, 3 mM $NaH_2PO_4$, and 10 mM Hepes, and 10 mM glucose). Fluorescence measurements were done in a Hitachi F-7000 or F-4500 fluorescence spectrophotometer using the Fura-2 conditions for excitation (340 and 380 nm) and emission (510 nm). The Fura-2 fluorescence response to $Ca^{2+}$ was calibrated from the ratio of 340/380 nm fluorescence values after subtraction of the background fluorescence of the cells at 340 and 380 nm as previously described (*Grynkiewicz et al., 1985*). The $Ca^{2+}$ release rate was defined as the change in $Ca^{2+}$ concentration during the initial 20 s after reagent addition. ΔF was calculated as the difference between the highest $Ca^{2+}$ peak and basal $Ca^{2+}$ levels, and recovery was defined as the change in $Ca^{2+}$ concentration after the peak was reached, measured over the indicated time intervals.

## Endoplasmic reticulum $Ca^{2+}$ measurements in permeabilized *T. gondii* tachyzoites

Tachyzoites freshly egressed and washed as described above were resuspend to a final density of $1\times10^9$ cells/ml in HBS buffer (135 mM NaCl, 5.9 mM KCl, 1.2 mM $MgCl_2$, 11.6 mM HEPES pH 7.3, 1.5 mM $CaCl_2$, 11.5 mM glucose) containing 1 mg/ml BSA, 0.2 mg/ml of pluronic F127 and 20 µM Mag-Fluo4-AM. The suspension was incubated at RT with mild shaking for 1 h, in the dark. Subsequently, parasites were washed two times and centrifuged at 5,000 rpm for 2 min to remove extracellular dye. The pellet was resuspended in 1.8 ml of CLM buffer (20 mM NaCl, 140 mM KCl, 20 mM PIPES, pH 7.0) containing 1 mM EGTA at $1\times10^9$ cells/ml. Parasites were permeabilized with 44.4 µg/ml digitonin for 6 min, washed twice with CLM containing 1 mM EGTA and cetrifuged at 5,000 rpm for 2 min to remove digitonin, then resuspended to a final density of $1\times10^9$ tachyzoites/ml and kept on ice. For each test, 50 µl ($5\times10^7$) of parasite suspension was added to 1.95 ml of CLM containing 1 mM EGTA and 0.375 mM $CaCl_2$ which results in 220 nM free $Ca^{2+}$ as calculated with MaxChelator. Fluorescence was measured with a Hitachi F-7000 or F-4500 fluorescence spectrophotometer (Excitation at 485 nm and emission at 520 nm). Ratio ($\Delta F/F_0/s$) was evaluated by measuring the rate of change in fluorescence over 20 s after reagent addition.

## Strain construction and maintenance

The organelle targeting of GCaMP6f was made by overlapping PCR. The N-terminal mitochondrial targeting sequence of the *T. gondii* SOD2 gene (*Pino et al., 2007*) was used to target GCaMP6f to the mitochondrion. The *GCaMP6f* gene for this construct was amplified by primers 9 and 10 (*Supplementary file 1*). After gel purification of the GCaMP6f and SOD2 sequences, the mitochondria targeting construct was built by overlapping PCR with the purified PCR products as template. This construct was then cloned into the Topo-blunt vector. After the sequence was verified by sequencing, the *SOD2-GCaMP6f* fragment was removed by BglII and AvrII digestion and cloned into the same restriction sites of the pDT7S4H3 (*Sheiner et al., 2011*) and pCTH3 (*Vella et al., 2020*) vectors. The pDT7S4H3-SOD2-GCaMP6f construct was introduced into RH parasites by electroporation. After selection with pyrimethamine, the parasites were sorted by FACS and then subcloned. Clones were selected based on the dynamic range of the response to ionomycin. The pCTH3-SOD2-GCaMP6f was introduced into the *iΔTgSERCA* mutant by electroporation. After selection with chloramphenicol, the parasites were sorted by FACS and then subcloned. The expression of GCaMP6f was verified by live-cell imaging, and western blots. The clone with the largest dynamic range, as evaluated using Ionomycin, was selected for further experiments. *Figure 5—figure supplement 1* shows live fluorescence confirming mitochondrial localization and fluorescence traces showing the response of whole parasites expressing GCaMP6f to the addition of $Ca^{2+}$ and Thapsigargin.

## GCaMP6f fluorescence measurements

Measurements with permeabilized parasites: *T. gondii* tachyzoites expressing SOD2-GCaMP6f were collected and washed two times at 5,000 rpm for 2 min with BAG. The parasite pellet was resuspended in 1.8 ml of BAG buffer containing 0.1 mM EGTA at $1 \times 10^9$ cells/ml. Permeabilization with 44.4 µg/ml digitonin for 6 min was done by following the fluorescence of GCaMP6f. Parasites were washed twice with the same buffer and centrifuged at 5,000 rpm for 2 min to remove digitonin, resuspended to a final density of $1 \times 10^9$ parasites/ml in intracellular buffer (140 mM Kgluconate, 10 mM NaCl, 2.7 mM $MgSO_4$, 200 µM EGTA, 65 µM $CaCl_2$, 10 mM HEPES, 10 mM Tris, pH 7.3, 1 mM Glucose) and kept on ice. 50 µl ($5 \times 10^7$) of the parasite suspension was mixed with 1.95 ml intracellular buffer for measurement. Measurements were done in a Hitachi 7000 fluorescence spectrophotometer set at 485 nm excitation and 509 nm emission. The uptake rate ($\Delta F/F_0/s$) was evaluated by measuring the % of change in fluorescence per second during the initial 20 s after reagent addition.

For measurements with intact parasites, they were collected, washed, and resuspended in BAG at $1 \times 10^9$ cells/ml for testing. 50 µl ($5 \times 10^7$) of the parasite suspension was mixed with 1.95 ml BAG containing 0.1 mM EGTA for measurement. The ratio ($\Delta F/F_0$) was evaluated by measuring the maximum change in fluorescence over 20 s after reagent addition (linear rate).

## Growth, invasion, and egress assays

Red-green invasion assays were performed as originally described (*Kafsack et al., 2004*), modified (*Chasen et al., 2017*) and adapted to use td-RFP-expressing parasites. The number of tachyzoites used was $2 \times 10^7$, and invasion was for 5 min. Plaque assays were performed as previously described (*Roos et al., 1994*) with modifications (*Liu et al., 2014*). 125 tachyzoites were used for infection of confluent six-well plates with hTERT fibroblasts, followed by an incubation time of 10 days prior to fixing and staining with crystal violet.

For egress assays, the monolayers of hTERT cells grown in 35 mm Mattek dishes were infected with 50,000 tdTomato-expressing parasites for 24 or 48 hr. Parasitophorous vacuoles containing 4–8 parasites were observed by microscopy after washing twice with Ringer's buffer without calcium. Dishes were filled with 1 ml of Ringer's buffer supplemented with either 100 µM EGTA or 1.8 mM $CaCl_2$. Images were collected in time-lapse mode with an acquisition rate of 3 s for 15 min. We observed that most of the *iΔTgSERCA* cells +/-ATc were still able to egress when stimulated with 1 or 0.5 µM ionomycin added 2 min after the start of the recordings with either 100 µM EGTA or 1.8 mM $CaCl_2$. We next tested lower concentrations of ionomycin (100 nM and 50 nM) in Ringer's buffer containing 1.8 mM $CaCl_2$. Egress was also triggered by adding 0.01% Saponin in the presence of 1.8 mM $CaCl_2$. For egress triggered by ionomycin, the percentage of vacuoles egressed after adding ionomycin during the 15 min of the video (2 min baseline + 13 min after adding ionomycin) was quantified

(from 100 vacuoles). For egress triggered by saponin, the time to egress after adding saponin was quantified.

For natural egress, the *iΔTgSERCA* mutant expressing td-tomato RFP was used to infect confluent hTERT cell monolayers 36 hr before adding ATc and 1 µM compound 1 (pyrrole 4-[2-(4-fluorophenyl)–5-(1-methylpiperidine-4-yl)–1H-pyrrol-3-yl]pyridine) (Cpd1) (*Donald et al., 2002*) dissolved in ethanol and the culture continued for 24 hr. After treating with Cpd1 for 24 hr, cultures showed intact vacuoles, which differed from the vehicle-treated plates (36 hr cultures plus ATc treatment for 24 hr or 48 hr without ATc), which were fully lysed. Following synchronization, the Cpd1-containing media was removed, and the vacuoles were washed twice with warm media lacking Cpd1. Fresh media without Cpd1 was added, and the plates were transferred to a prewarmed DeltaVision microscope stage set to 37 °C. After 10 min at 37 °C, egress of the full vacuoles was enumerated. We counted each plate for 1 min and evaluated at least 100 vacuoles per experiment. Three independent biological experiments were conducted and summarized.

For replication assays, hTERT cells were grown on 35 mm MatTek dishes. Each dish was infected with 50,000 tdTomato-expressing parasites. 24 hr after the infection, the number of parasites per PV was counted using a fluorescence microscope. For each experiment, at least 100 PVs were counted. Results were the average of three independent experiments (*Li et al., 2021*).

## Microscopy and western blot analyses

Tachyzoites were grown on hTERT cells on cover slips for ~24 hr, washed twice with BAG and fixed with 4% formaldehyde for 1 hr, followed by permeabilization with 0.3% Triton X-100 for 20 min, and blocking with 3% bovine serum albumin. IFAs were performed as previously described (*Miranda et al., 2010*). Fluorescence images were collected with an Olympus IX-71 inverted fluorescence microscope with a Photometrix CoolSnapHQ CCD camera driven by DeltaVision software (Applied Precision, Seattle, WA). Super-resolution microscopy was performed using a Zeiss ELYRA S1 (SR-SIM) system mounted on a high-resolution Axio Observer Z1 inverted microscope. The setup included transmitted light (HAL), UV (HBO), and high-power solid-state laser illumination sources (405/488/561 nm), a 100× oil immersion objective, and an Andor iXon EM-CCD camera. Image acquisition and structured illumination analysis were conducted using ZEN software (Zeiss) with the SIM analysis module. Rat anti-HA antibody (Roche) was used at a 1:25 dilution, and mouse anti-HA antibody (Covance) was used at a 1:200 dilution. Affinity-purified guinea pig anti-TgSERCA antibody was used at a 1:500 dilution.

Western blot analysis was performed as previously described (*Liu et al., 2014*). Rat anti-HA antibody from Roche was used at a dilution of 1:200. Mouse anti-HA antibody from Covance was used at a dilution of 1:1,000. The guinea pig anti-TgSERCA antibody was used at a dilution of 1:2000. Secondary goat anti-rat or mouse antibody conjugated with HRP was used at 1:5,000. Mouse anti-α-tubulin at a dilution of 1:5,000 was used for loading control.

## Transmission electron microscopy

For ultrastructural observations of intracellular *T. gondii* by thin-section transmission EM, infected human foreskin fibroblast cells were fixed in 2.5% glutaraldehyde in 0.1 mM sodium cacodylate (EMS) and processed as described (*Coppens and Joiner, 2003*). Ultrathin sections of infected host cells were stained before examination with a Hitachi 7600 EM under 80 kV. For quantitative measurement of distance between organelles, the closest point between *T. gondii's* organelles and ER membrane was measured using ImageJ and was performed on 47 representative electron micrographs at the same magnification for accurate comparison between organelles.

## Statistical analysis

Statistical analyses were performed by Student's t-test using GraphPad PRISM version 9. Error bars shown represent mean ±SD (standard deviation) of at least three independent biological replicates. Unpaired two-tailed t-test performed in all comparisons.

## Acknowledgements

The authors thank Dr. Muthugapatti Kandasamy and the Biomedical Microscopy Core of the University of Georgia for the use of the microscopes. The CTEGD Cytometry Shared Resource Laboratory provided access and training to state-of-the-art flow cytometry analyzers. We would like to thank

David Sibley for the generous gift of the mouse SERCA antibody and the plasmid for SERCA expression and Vern Carruthers for the anti-CPL antibody. This work was supported by the U.S. National Institutes of Health grants AI128356, AI154931 and AI174600 to SNJM, and R01AI166921 to IC.

## Additional information

### Funding

| Funder | Grant reference number | Author |
| --- | --- | --- |
| National Institutes of Health | R01AI128356 | Silvia NJ Moreno |
| National Institutes of Health | R01AI174600 | Silvia NJ Moreno |
| National Institutes of Health | R21AI154931 | Silvia NJ Moreno |
| National Institutes of Health | R01AI166921 | Isabelle Coppens |

The funders had no role in study design, data collection and interpretation, or the decision to submit the work for publication.

### Author contributions

Zhu-Hong Li, Data curation, Formal analysis, Investigation, Methodology, Writing – review and editing; Beejan Asady, Formal analysis, Investigation, Methodology, Writing - original draft; Le Chang, Catherine Li, Investigation, Methodology; Myriam Andrea Hortua Triana, Data curation, Investigation, Methodology, Writing – review and editing; Isabelle Coppens, Data curation, Investigation, Methodology; Silvia NJ Moreno, Conceptualization, Resources, Supervision, Funding acquisition, Writing - original draft, Project administration, Writing – review and editing

### Author ORCIDs

Zhu-Hong Li ⬤ https://orcid.org/0000-0003-4940-0729
Myriam Andrea Hortua Triana ⬤ https://orcid.org/0000-0001-5964-8512
Silvia NJ Moreno ⬤ https://orcid.org/0000-0002-2041-6295

### Ethics

We generated antibodies in guinea pigs. Animals were handled according to our approved institutional animal care and use committee (IACUC) protocols (A2021 03-005-A5) of the University of Georgia.

Reviewer #1 (Public review): https://doi.org/10.7554/eLife.101894.3.sa1
Author response https://doi.org/10.7554/eLife.101894.3.sa2

## Additional files

### Supplementary files

MDAR checklist
Supplementary file 1. Primers used in this study.

### Data availability

All data generated or analyzed during this study are included in the manuscript and supporting files. Source data files have been provided for Figures 1-5.

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
