## [Editor Report · eLife Assessment]

This **important** study shows that calcium stores in the endoplasmic reticulum of the parasitic protozoan, *Toxoplasma gondii* play a major role in regulating calcium levels in the cytosol as well as other organelles such as the mitochondrion. Advanced imaging techniques, including use of genetically encoded calcium indicators provide **compelling** evidence for the role of the SERCA-Ca^2+^-ATPase pump in regulating organellar calcium levels. However, it remains unclear whether inter-organellar calcium transport occurs via ER-mitochondria membrane contact sites or other mechanisms. This work will be of interest to cell and molecular biologists interested in calcium signalling in divergent eukaryotes.

---

## [Referee Report · Reviewer #1 (Public review)]

Li et al. investigate Ca2+ signaling in *T. gondii* and argue that Ca2+ tunnels through the ER to other organelles to fuel multiple aspects of *T. gondii* biology. They focus in particular on TgSERCA as the presumed primary mechanism for ER Ca2+ filling. Although, when TgSERCA was knocked out there was still a Ca2+ release in response to TG present. Overall the data supports a model where the Ca2+ filling state of the ER modulates Ca2+ dynamics in other organelles.

Comments on revisions:

I thank the authors for their careful revisions and response to my comments, which have been addressed.

Regarding the most critical point of the paper that is Ca2+ transfer from the ER to other organelles, the authors in their rebuttal and in the revised manuscript argue that ER Ca2+ is critical to redistribute and replenish Ca2+ in other organelles in the cell. I agree this conclusion and think it is best stated in the authors' response to point #7: "We propose that this leaked calcium is subsequently taken up by other intracellular compartments. This effect is observed immediately upon TG addition. However, pre-incubation with TG or knockdown of SERCA reduces calcium storage in the ER, thereby diminishing the transfer of calcium to other stores."

In their rebuttal the authors particularly highlight experiments in Figures 1H-K, 4G-H, and 5H-K in support of this conclusion. The data in Fig 1H-K show that with TG there is increased Ca2+ release from acidic stores. In all cases TG results in a rise in cytoplasmic Ca2+ that could load the acidic stores. So under those conditions the increased acidic organelle Ca2+ is likely due to a preceding high cytosolic Ca2+ transient due to TG. The experiments in 4G-H and 5H-K are more convincing and supportive of an important role of ER Ca2+ to maintain Ca2+ levels in other organelles. Overall, and to avoid a detailed, lengthy discussion of every point, the data support a model where in the absence of SERCA activity ER Ca2+ is reduced as well as Ca2+ in other organelles. I think it would be helpful to present and discuss this finding throughout the manuscript as under physiological conditions ER Ca2+ is regularly mobilized for signaling and homeostasis and this maintains Ca2+ levels in other organelles. This is supported by the new experiment in Supp Fig. 2A.

---

## [Author Response]

The following is the authors’ response to the original reviews

**Public Reviews:**

**Reviewer #1 (Public review):**
Li et al. investigate Ca2+ signaling in *T. gondii* and argue that Ca2+ tunnels through the ER to other organelles to fuel multiple aspects of *T. gondii* biology. They focus in particular on TgSERCA as the presumed primary mechanism for ER Ca2+ filling. Although, when TgSERCA was knocked out there was still a Ca2+ release in response to TG present.

Note that we did not generate a complete SERCA knockout, as this gene is essential, and its complete loss would not permit the isolation of viable parasites. Instead, we created conditional mutants that downregulate the expression of SERCA. Importantly, some residual activity is present in the mutant after 24 h of ATc treatment as shown in Fig 4C. This is consistent with our Western blots, which demonstrate the presence of residual SERCA protein at 1, 1.5 and 2 days post ATc treatment (Fig. 3B). We have clarified this point in the revised manuscript (lines 232233). See also lines 97-102.

Overall the Ca2+ signaling data do not support the conclusion of Ca2+ tunneling through the ER to other organelles in fact they argue for direct Ca2+ uptake from the cytosol. The authors show EM membrane contact sites between the ER and other organelles, so Ca2+ released by the ER could presumably be taken up by other organelles but that is not ER Ca2+ tunneling. They clearly show that SERCA is required for *T. gondii* function.

Overall, the data presented to not fully support the conclusions reached

We agree that the data does not support Ca^2+^ tunneling as defined and characterized in mammalian cells. In response to this comment, we have modified the title and the text accordingly.

However, we respectfully would like to emphasize that the study demonstrates more than just the role of SERCA in *T. gondii* “function”. Our findings reveal that the ER, through SERCA activity, sequesters calcium following influx through the PM (see reviewer 2 comment). The ER calcium pool is important for replenishing other intracellular compartments.

The experiments support a model in which the ER actively takes up cytosolic Ca²⁺ as it enters the parasite and contributes to intracellular Ca²⁺ redistribution during transitions between distinct extracellular calcium environments. We believe that the role of the ER in modulating intracellular calcium dynamics is demonstrated in Figures 1H–K, 4G-H, and 5H–K. To highlight the relevance of these findings, we have included an expanded discussion in the revised manuscript. See lines 443-449 and 510-522.

Data argue for direct Ca2+ uptake from the cytosol

The ER most likely takes up calcium from the cytosol following its entry through the PM and redistributes it to the other organelles. We deleted any mention of the word “tunneling” and replaced it with transfer and re-distribution as they reflect our experimental findings more accurately.

We interpret the experiments shown in Figure 1 H and I as re-distribution because the amount of calcium released after nigericin or GPN are greatly enhanced after TG addition. We first add calcium to allow intracellular stores to become filled, followed by the addition of TG, which allows calcium leakage from the ER. This leaked calcium can either enter the cytosol and be pumped out or be taken up by other organelles. Our interpretation is that this process leads to an increased calcium content in acidic compartments.

We conducted an additional experiment in which SERCA was inhibited prior to calcium addition, allowing cytosolic calcium to be exported or taken up by acidic stores. We observed a change in the GPN response (Fig. S2A), possibly indicating that the PLVAC can sequester calcium when SERCA is inactive. While this may support the reviewer’s view, TG treatment does not reflect physiological conditions and may enhance calcium transfer to other compartments. Although the result is interesting, interpretation is complicated by the use of parasites in suspension and drug exposure in solution. Single-parasite measurements are not feasible due to weak signals, and adhered parasites are even less physiological than those in suspension.

In support of our view, the experiments shown in Figs 4G and H show that down regulating SERCA reduces significantly the response to GPN indicating diminished acidic store loading. In Fig 5I we observe that mitochondrial calcium uptake is reduced in the iDSERCA (+ATc) mutant in response to GPN. Fig 2B demonstrates that TgSERCA can take up calcium at 55 nM, close to resting cytosolic calcium while in Figures 5E and S5B we show that the mitochondrion is not responsive to an increase of cytosolic calcium. Uptake by the mitochondria requires much higher concentrations (Fig 5B-C), which may be achieved within microdomains at MCS between the ER and mitochondrion. This is also consistent with findings reported by Li et al (Nat Commun. 2021) where similar microdomains mediated transfer of calcium to the apicoplast (Fig. 7 E and F of the mentioned reference) was observed.

**Reviewer 2 (Public review):**
The role of the endoplasmic reticulum (ER) calcium pump TgSERCA in sequestering and redistributing calcium to other intracellular organelles following influx at the plasma membrane.*T. gondii* transitions through life cycle stages within and exterior to the host cells, with very different exposures to calcium, adds significance to the current investigation of the role of the ER in redistributing calcium following exposure to physiological levels of extracellular calciumThey also use a conditional knockout of TgSERCA to investigate its role in ER calcium store-filling and the ability of other subcellular organelles to sequester and release calcium. These knockout experiments provide important evidence that ER calcium uptake plays a significant role in maintaining the filling state of other intracellular compartments.

We thank the reviewer.

While it is clearly demonstrated, and not surprising, that the addition of 1.8 mM extracellular CaCl2 to intact *T. gondii* parasites preincubated with EGTA leads to an increase in cytosolic calcium and subsequent enhanced loading of the ER and other intracellular compartments, there is a caveat to the quantitation of these increases in calcium loading. The authors rely on the amplitude of cytosolic free calcium increases in response to thapsigargin, GPN, nigericin, and CCCP, all measured with fura2. This likely overestimates the changes in calcium pool sizes because the buffering of free calcium in the cytosol is nonlinear, and fura2 (with a Kd of 100-200 nM) is a substantial, if not predominant, cytosolic calcium buffer. Indeed, the increases in signal noise at higher cytosolic calcium levels (e.g. peak calcium in Figure 1C) are indicative of fura2 ratio calculations approaching saturation of the indicator dye.

We acknowledge the limitations associated with using Fura-2 for cytosolic calcium measurements. However, according to the literature (Grynkiewicz, Get al. (1985). J. Biol. Chem. 260 (6): 3440–3450. PMID 3838314) Fura-2 is suited for measurements between 100 nM and 1 µM calcium. The responses in our experiments were within that range and the experiments with the SERCA mutant and mitochondrial GCaMPfs supports the conclusions of our work.

However, we agree with the reviewer that the experiment shown in Fig 1C (now Fig 1D) presents a response that approaches the limit of the linear range of Fura-2. In response to this, we have replaced this panel with a more representative experiment that remains within the linear range of the indicator (revised Fig 1D). Additionally, we have included new experiments adding GPN along with corresponding quantifications, which further support our conclusions regarding calcium dynamics in the parasite.

Another caveat, not addressed, is that loading of fura2/AM can result in compartmentalized fura2, which might modify free calcium levels and calcium storage capacity in intracellular organelles.

We are aware of the potential issue of Fura-2 compartmentalization, and our protocol was designed to minimize this effect. We load cells with Fura-2 for 26 min at room temperature, then maintain them on ice, and restrict the use of loaded parasites to 2-3 hours. We have observed evidence of compartmentalization as this is reflected in increasing concentrations of resting calcium with time. We carry out experiments within a time frame in which the resting calcium stays within the 100 nM range. We have included a sentence in the Materials and Methods section. Lines 604-606.

Additionally, following this reviewer’s suggestion, we performed further experiments to directly assess compartmentalization. See below the full response to reviewer 2.

The finding that the SERCA inhibitor cyclopiazonic acid (CPA) only mobilizes a fraction of the thapsigargin-sensitive calcium stores in *T. gondii* coincides with previously published work in another apicomplexan parasite, *P. falciparum*, showing that thapsigargin mobilizes calcium from both CPA-sensitive and CPA-insensitive calcium pools (Borges-Pereira et al., 2020, DOI: 10.1074/jbc.RA120.014906). It would be valuable to determine whether this reflects the off-target effects of thapsigargin or the differential sensitivity of TgSERCA to the two inhibitors.

This is an interesting observation, and we now include a discussion of this result considering the Plasmodium study and include the citation. Lines 436-442.

Figure S1 suggests differential sensitivity, and it shows that thapsigargin mobilizes calcium from both CPA-sensitive and CPA-insensitive calcium pools in *T. gondii*. Also important is that we used 1 µM TG as we are aware that TG has shown off-target effects at higher concentrations. TG is a well-characterized, irreversible SERCA inhibitor that ensures complete and sustained inhibition of SERCA activity. In contrast, CPA is a reversible inhibitor whose effectiveness is influenced by ATP levels, and it may only partially inhibit SERCA or dissociate over time, allowing residual Ca²⁺ reuptake into the ER.

Additionally, as suggested by the reviewer we performed experiments using the Mag-Fluo-4 protocol to compare the inhibitory effects of CPA and TG. These results are presented in Fig. S3 (Lines 217-223). Under the conditions of the Mag-Fluo-4 assay with digitonin-permeabilized cells, both TG and CPA showed similar rates of Ca^2+^ leakage following the addition of the inhibitor. This may indicate that under the conditions of the Mag-Fluo-4 experiments the rate of Ca^2+^ leak is mostly determined by the intrinsic leak mechanism and not by the nature of the inhibitor. By contrast, in intact Fura-2–loaded cells, CPA induces a smaller cytosolic Ca²⁺ increase than TG, consistent with less efficient SERCA inhibition likely due to its reversibility and possibly incomplete inhibition under cellular conditions.

The authors interpret the residual calcium mobilization response to Zaprinast observed after ATc knockdown of TgSERCA (Figures 4E, 4F) as indicative of a target calcium pool in addition to the ER. While this may well be correct, it appears from the description of this experiment that it was carried out using the same conditions as Figure 4A where TgSERCA activity was only reduced by about 50%.

We partially agree with the reviewer that 50% knockdown of TgSERCA means that the ER may still be targeted by zaprinast, and that there is no definitive evidence of the involvement of another calcium pool. The Mag-Fluo-4 experiment, while we acknowledge that the fluorescence of MagFluo-4 is not linear to calcium, indicates that SERCA activity is present even after 24 hr of ATc treatment. However, when Zaprinast is added after TG, we observed a significant calcium release in wild type cells. This result suggests the presence of another large calcium pool than the one mobilized by TG (PMID: 2693306).

We recently published work describing the Golgi as a calcium store in Toxoplasma (PMID: 40043955) and we showed in Fig. S4 D-G of that work, that GPN treatment of tachyzoites loaded with Fura-2 diminished the Zaprinast response indicating that they could be impacting a similar store. In the present study we performed additional experiments in which TG was followed by GPN and Zaprinast showing a similar pattern. GPN significantly diminished the Zaprinast response. These results are shown now in Figure S2B. We address these possibilities in the discussion and interpretation of the result. Lines 451-460.

The data in Figures 4A vs 4G and Figures 4B vs 4H indicate that the size of the response to GPN is similar to that with thapsigargin in both the presence and absence of extracellular calcium. This raises the question of whether GPN is only releasing calcium from acidic compartments or whether it acts on the ER calcium stores, as previously suggested by Atakpa et al. 2019 DOI: 10.1242/jcs.223883. Nonetheless, Figure 1H shows that there is a robust calcium response to GPN after the addition of thapsigargin.

The results of the indicated experiments did not exclude the possibility that GPN can also mobilize some calcium from the ER besides acidic organelles. We don’t have any evidence to support that GPN can mobilize calcium from the ER either. Based on our unpublished work, we think GPN mainly release calcium from the PLVAC. We included the mentioned citation and discuss the result considering the possibility that GPN may be acting on more than one store. Lines 451-460.

An important advance in the current work is the use of state-of-the-art approaches with targeted genetically encoded calcium indicators (GECIs) to monitor calcium in important subcellular compartments. The authors have previously done this with the apicoplast, but now add the mitochondria to their repertoire. Despite the absence of a canonical mitochondrial calcium uniporter (MCU) in the Toxoplasma genome, the authors demonstrate the ability of *T. gondii* mitochondrial to accumulate calcium, albeit at high calcium concentrations. Although the calcium concentrations here are higher than needed for mammalian mitochondrial calcium uptake, there too calcium uptake requires calcium levels higher than those typically attained in the bulk cytosolic compartment. And just like in mammalian mitochondria, the current work shows that ER calcium release can elicit mitochondrial calcium loading even when other sources of elevated cytosolic calcium are ineffective, suggesting a role for ER-mitochondrial membrane contact sites. With these new tools in hand, it will be of great value to elucidate the bioenergetics and transport pathways associated with mitochondrial calcium accumulation in *T. gondii*.

We thank this reviewer praising our work. Studies of bioenergetics and transport pathways associated with mitochondrial calcium accumulation is part of our future plans mentioned in lines 520-522 and 545.

The current studies of calcium pools and their interactions with the ER and dependence on SERCA activity in T. gondi are complemented by super-resolution microscopy and electron microscopy that do indeed demonstrate the presence of close appositions between the ER and other organelles (see also videos). Thus, the work presented provides good evidence for the ER acting as the orchestrating organelle delivering calcium to other subcellular compartments through contact sites in T. gondi, as has become increasingly clear from work in other organisms.

Thank you.

**Reviewer #3 (Public review):**
This manuscript describes an investigation of how intracellular calcium stores are regulated and provides evidence that is in line with the role of the SERCA-Ca2+ATPase in this important homeostasis pathway. Calcium uptake by mitochondria is further investigated and the authors suggest that ER-mitochondria membrane contact sites may be involved in mediating this, as demonstrated in other organisms.The significance of the findings is in shedding light on key elements within the mechanism of calcium storage and regulation/homeostasis in the medically important parasite *Toxoplasma gondii* whose ability to infect and cause disease critically relies on calcium signalling. An important strength is that despite its importance, calcium homeostasis in Toxoplasma is understudied and not well understood.

We agree with the reviewer. Thank you.

A difficulty in the field, and a weakness of the work, is that following calcium in the cell is technically challenging and thus requires reliance on artificial conditions. In this context, the main weakness of the manuscript is the extrapolation of data. The language used could be more careful, especially considering that the way to measure the ER calcium is highly artificial - for example utilising permeabilization and over-loading the experiment with calcium. Measures are also indirect - for example, when the response to ionomycin treatment was not fully in line with the suggested model the authors hypothesise that the result is likely affected by other storage, but there is no direct support for that.

The Mag-Fluo-4-based protocol for measuring intraluminal calcium is well established and has been extensively used in mammalian cells, DT40 cells and other cells for measuring intraluminal calcium, activity of SERCA and response to IP3 (Some examples: PMID: 32179239, PMID: 15963563, PMID: 19668195, PMID: 30185837, PMID: 19920131).

Furthermore, we have successfully employed this protocol in previous work, including the characterization of the *Trypanosoma brucei* IP3R (PMID: 23319604) and the assessment of SERCA activity in Toxoplasma (PMID: 40043955 and 34608145). The citation PMID: 32179239 provides a detailed description of the protocol, including references to its prior use. In addition, the schematic at the top of Figure 2 summarizes the experimental workflow, reinforcing that the protocol follows established methodologies. We included more references and an expanded discussion, lines 425-435.

We respectfully disagree with the concern regarding potential calcium overloading. The cells used in our assays were permeabilized, which is a critical step that allows to precisely control calcium concentrations. All experiments were conducted at 220 nM free calcium, a concentration within the physiological range of cytosolic calcium fluctuations. This concentration was consistently used across all studies described above. Importantly, permeabilization ensures that the dye present in the cytosol becomes diluted, and allows MgATP (which cannot cross intact membranes) to access the ER membrane, in addition to be able to expose the ER to precise calcium concentrations.

The Mag-Fluo-4 loading conditions are designed to allow compartmentalization of the indicator to all intracellular compartments and the calcium uptake stimulated by MgATP exclusively occurs in the compartment occupied by SERCA as only SERCA is responsive to MgATP-dependent transport in this experimental setup.

Regarding the use of IO, we would like to clarify that its broad-spectrum activity is welldocumented. As a calcium ionophore, IO facilitates calcium release across multiple membranes, and not just the ER leading to a more substantial calcium release compared to the more selective effect of TG. The results observed with IO were consistent with this expected broader activity and support our interpretation.

Lastly, we emphasize that the experiment in Figure 2 was designed specifically to assess SERCA activity *in situ* under defined conditions. It was not intended to provide a comprehensive characterization of the role of TgSERCA in the parasite. We now clarify this distinction in the revised Discussion lines 425-435.

Below we provide some suggestions to improve controls, however, even with those included, we would still be in favour of revising the language and trying to avoid making strong and definitive conclusions. For example, in the discussion perhaps replace "showed" with "provide evidence that are consistent with..."; replace or remove words like "efficiently" and "impressive"; revise the definitive language used in the last few lines of the abstract (lines 13-17); etc. Importantly we recommend reconsidering whether the data is sufficiently direct and unambiguous to justify the model proposed in Figure 7 (we are in favour of removing this figure at this early point of our understanding of the calcium dynamic between organelles in Toxoplasma).

We thank the reviewer for the suggestions and we modified the language as suggested. We limited the use of the word "showed" to references to previously published work. We deleted the other words.

Figure 7 is intended as a conceptual model to summarize our proposed pathways, and, like all models, it represents a working hypothesis that may not fully capture the complexity of calcium dynamics in the parasite. In light of the reviewer’s comments, we revised the figure and legend to clearly distinguish between pathways for which there is experimental evidence from those that are hypothetical.

Another important weakness is poor referencing of previous work in the field. Lines 248250 read almost as if the authors originally hypothesised the idea that calcium is shuttled between ER and mitochondria via membrane contact sites (MCS) - but there is extensive literature on other eukaryotes which should be first cited and discussed in this context. Likewise, the discussion of MCS in Toxoplasma does not include the body of work already published on this parasite by several groups. It is informative to discuss observations in light of what is already known.

The sentence in which we state the hypothesis about the calcium transfer refers specifically to Toxoplasma. To clarify this, we have now added the phrase “In mammalian cells” (Line 311) and included additional citations, as suggested by the reviewer. While only a few studies have described membrane contact sites (MCSs) in Toxoplasma, we do cite several pertinent articles (e.g., lines 479-486). We believe that we cited all articles mentioning MCS in *T. gondii.*

However, we must clarify to the reviewer that the primary focus of our study is not to characterize or confirm the presence of MCSs in *T. gondii*, but rather to demonstrate functional calcium transfer between the ER and mitochondria. Our data support the conclusion that this transfer requires close apposition of these organelles, consistent with the presence of MCSs.

**Recommendations for the authors:**

**Reviewer #1 (Recommendations for the authors):**
(1) Line 45: change influx to release as Ca2+ influx usually referred to Ca2+ entry from the extracellular space. Same for line 71.

Corrected, line 47 and 73

(2) Line 54: consider toning down the strong statement of 'widely' accepted as ER Ca2+ subdomain heterogeneity remains somewhat debated.

Changed the sentence to “it has been proposed”, Line 56

(3) Line 119-21: A lower release in response to TG is typical and does not reflect TG specific for SERCA. It is due to the slow kinetics of Ca2+ leak out of the ER allowing other buffering and transport mechanisms to act. Also, could be a reflection of the duration after TG treatment to allow complete store depletion. Figure S1A-B shows that there is still Ca2+ in the stores following TG but the TG signal does not go back to baseline arguing that the leak is still active. Hence the current data does not address the specificity of TG for TgSERCA. Please revise the statement accordingly.

Thank for the suggestion, we changed the sentence to this: “This result could reflect the slow kinetics of Ca²⁺ leak from the ER, allowing other buffering and transport mechanisms to mitigate the phenomenon. Alternatively, it may indicate the duration after TG treatment allowing time to complete store depletion. As shown in Figure S1A-B, residual Ca²⁺ remains in the stores after TG treatment, and the TG-induced phenomenon does not return to baseline, suggesting that the leak remains active”. Lines 124-128

(4) Figure 1C: the authors interpret the data 'This Ca2+ influx appeared to be immediately taken up by the ER as the response to TG was much greater in parasites previously exposed to extracellular Ca2+'. I don't understand this interpretation, in Ca2+-containing solution it would expected to have a larger signal as TG is likely to activate store-operated Ca2+ entry which would contribute to a larger cytosolic Ca2+ transient. Does *T. gondii* have SOCE? It cannot be uptake into the ER as SERCA is blocked. Unless the authors are arguing for another ER Ca2+ uptake pathway? But why are Ca2+ uptake in the ER would lower the signal whereas the data show an increased signal?

We pre-incubated the suspension with calcium to allow filling of the stores, while SERCA is still active, and added thapsigargin (TG) at 400 seconds to measure calcium release. The experiment was designed to introduce the concept that the ER may have access to extracellular calcium, a phenomenon not yet clearly demonstrated in Toxoplasma. We did not expect to have less release by TG but if the ER is not efficient in filling after extracellular calcium entry it would be expected to have a similar response to TG. Yes, it is very possible that when we add TG we are also seeing more calcium entry through the PM as we previously proposed that the increased cytosolic Ca^2+^ may regulate Ca^2+^ entry. However, the evidence does not support that this increased entry would be triggered by store depletion. The experiments with the SERCA mutant (Fig. 4D) shows that in the conditional knockout mutant, the ER is partially depleted, yet this does not lead to enhanced calcium entry, suggesting that the depletion alone is not sufficient to trigger increased influx.

There is no experimental evidence supporting the regulation of calcium entry by store depletion in Toxoplasma (PMID: 24867952). We revised the text to clarify this point and expanded the discussion on store-operated calcium entry (SOCE). While it is possible that a channel similar to Orai exists in Toxoplasma, it is highly unlikely to be regulated by store depletion, as there is no gene homologous to STIM. If store-regulated calcium entry does occur in Toxoplasma, it is likely mediated through a different, still unidentified, mechanism. Lines 461-467.

(5) The choice of adding Ca2+ first followed by TG is curious as it is more difficult to interpret. Would be more informative to add TG, allow the leak to complete, and then add Ca2+ which would allow temporal separation between Ca2+ release from stores and Ca2+ influx from the extracellular space. Was this experiment done? If not would be useful to have the data.

Yes, this experiment was already published: PMID: 24867952 and PMID: 38382669.

It mainly highlighted that increased cytosolic calcium may regulate calcium entry most likely through a TRP channel. See our response to point 4 and the description of the new Fig. S2 in the response to point 7.

(6) Line 136-39: these experiments as designed - partly because of the issues discussed above - do not address the ability of organelles to access extracellular Ca2+ or the state of refilling of intracellular Ca2+ stores. They can simply be interpreted as the different agents (TG, Nig, GPN, CCCP) inducing various levels of Ca2+ influx.

Concerning TG, the experiment shown in Fig. 4D shows that depletion of the ER calcium does not result in stimulation of calcium entry, indicating the absence of classical SOCE activation in Toxoplasma.

To our knowledge, neither mitochondria nor lysosomes (or other acidic compartments) are capable of triggering classical SOCE in mammalian cells.

Given that the ER in Toxoplasma lacks the canonical components required to initiate SOCE, it is unclear why the mitochondria or acidic compartments would be able to do so. While it is possible that *T. gondii* utilizes an alternative mechanism for store-operated calcium entry, investigating such a pathway would require a comprehensive study. In mammalian systems, it took almost 15 years and the efforts of multiple research groups to identify the molecular components of SOCE. Expecting this complex question to be resolved within the scope of a single study is unrealistic.

Our current data show that the mitochondrion is unable to access calcium from the cytosol, as shown in Figure 5E. Performing a similar experiment for the PLVAC would be ideal; however, expression of fluorescent calcium indicators in this organelle has not been successful. This is likely due to the presence of several proteases that degrade expressed proteins, as well as the acidic environment, which quenches fluorescence. These challenges have made studying calcium dynamics in the PLVAC particularly difficult.

To address the reviewer’s comment, we performed an additional experiment presented in Fig. S2A. In this experiment, we first inhibited SERCA with thapsigargin (TG), preventing calcium uptake into the ER, and subsequently added calcium to the suspension. Under these conditions, calcium cannot be sequestered by the ER. We then applied GPN and quantified the response, comparing it to a similar experimental condition without TG. Indeed, under these conditions, we observed a significant but modest increase in the GPN-induced response, suggesting that the PLVAC may be capable of directly taking up calcium from the cytosol. However, this occurs under conditions of SERCA inhibition which creates nonphysiological conditions with elevated cytosolic calcium levels and the presence of TG may promote additional ER leakage, both of which could artificially enhance PLVAC uptake. Under physiological conditions, with functional SERCA activity, the ER would likely sequester cytosolic calcium more efficiently, thereby limiting calcium availability for PLVAC direct uptake. Thus, while the result is intriguing, it may not reflect calcium handling under normal cellular conditions. See lines 172-178.

(7) Figure 1H-I: I disagree with the authors' interpretation of the results (lines 144-153). The data argue that by blocking ER Ca2+ uptake by TG, other organelles take up Ca2+ from the cytosol where it accumulates due to the leak and Ca2+ influx as is evident from the data allowing more release. The data does not argue for ER Ca2+ tunneling to other organelles. Tunneling would be reduced in the presence of TG (see PMID: 30046136, 24867608).

We partially agree with this concern. In our experiments, TG was used to inhibit SERCA and block calcium uptake into the ER, allowing calcium to leak into the cytosol. We propose that this leaked calcium is subsequently taken up by other intracellular compartments. This effect is observed immediately upon TG addition. However, pre-incubation with TG or knockdown of SERCA reduces calcium storage in the ER, thereby diminishing the transfer of calcium to other stores.

To further support our claim, we performed additional experiments in the absence of extracellular calcium, now presented in Figure 1J-K. We observed that calcium release triggered by GPN or nigericin was significantly enhanced when both agents were added after TG. These results suggest that calcium initially released from the ER can be sequestered by other compartments. As mentioned, we deleted any mention of “tunneling,” but we believe the data support the occurrence of calcium transfer. New results described in lines 166-171.

The experiment in Fig S2A described in the response to (6) also addresses this concern. Under physiological conditions with functional SERCA, cytosolic calcium would likely be rapidly sequestered by the ER, limiting its availability to other compartments. See lines 172178.

(8) Line 175: SERCA-dependent Ca2+ uptake is higher at 880 nM as would be expected yet the authors state that it's optimal at 220 nM Ca2+ ?

Yes, it is true that the SERCA-dependent Ca^2+^ uptake rate is higher at elevated Ca²⁺ concentrations. We chose to use 220 nM free calcium because of several reasons: (1) this concentration is close to physiological cytosolic levels fluctuations; (2) it is commonly used in studies of mammalian SERCA; and (3) calcium uptake is readily detectable at this level. While this may not represent the maximal activity conditions for SERCA, we believe it is a reasonable and physiologically relevant choice for assessing calcium transport activity SERCA-dependent. We added one sentence to the results explaining this reasoning (lines 204-207) and we deleted the word optimal.

(9) Figure 3H: the saponin egress data support the conclusion that organelles Ca2+ take up cytosolic Ca2+ directly without the need for ER tunneling.

The saponin concentration used permeabilizes the host cell membrane, allowing the intracellular tachyzoite to be surrounded with the added higher extracellular calcium concentration. The saponin concentration used does not affect the tachyzoite membrane as the parasite is still moving and calcium oscillations were clearly seen under similar conditions (PMID: 26374900). The resulting calcium increase in the tachyzoite cytosol is what stimulates parasite motility and egress. Since SERCA activity is reduced in the mutant, cytosolic calcium accumulates more rapidly, reaching the threshold for egress sooner and thereby accelerating parasite exit. The result does not support that the other stores contribute to this because of the Ionomycin response, which shows that egress is diminished in the mutant, likely because the calcium stores are depleted. We added an explanation in the results, lines 262-269 and the discussion, lines 532-539.

(10) Figure S2: the HA and SERCA signals do not match perfectly? Could this reflect issues with HA tagging, potentially off-target effects? Was this tested?

These are not off-target effects, as we did not observe them in the control cells lacking HA tagging. The HA signal also disappeared after treatment with ATc, further confirming that the IFA signal is specific. We agree with the reviewer that the signals do not align perfectly. This discrepancy could be due to differences in antibody accessibility or the fact that the two antibodies recognize different regions of the protein. We added a sentence about this in the result; lines 240-243.

**Reviewer #2 (Recommendations for the authors):**
The description of the data of Figures 1B and S1A starting on line 108 would be easier to follow if Figure S1A was actually incorporated into Figure 1. It is not clear why these two complementary experiments were separated since they are both equally important in understanding and interpreting the data.

We re-arranged figure 1 and incorporated S1A now as Fig 1C.

As noted in the public comments, loading of fura2/AM can result in compartmentalized fura2, which can contaminate the cytosolic calcium measurements and might modify free calcium levels and calcium storage capacity in intracellular organelles. This can be assessed using the digitonin permeabilization method used in the MagFluo4 measurements, but in this case, detecting the fura2 signal remaining after cell permeabilization.

As suggested by the reviewer, we measured Fura-2 compartmentalization by permeabilizing cells with digitonin as we do for the Mag-Fluo-4 and the fluorescence was reduced almost completely and was unresponsive to any additions (see Author response image 1).

**Author response image 1. sa2fig1:** *T. gondii* tachyzoites in suspension exposed to Thapsigargin Calcium and GPN. The dashed lines shows and experiments using the same conditions but parasites were permeabilized with digitonin shows a similar experiment with parasites exposed to MgATP.to release the cytosolic Fura. Part B

Following the public comment regarding the residual calcium mobilization response to Zaprinast observed after 24 h ATc knockdown of SERCA (Figsures 4E, 4F, as explained in the legend to Figure 4), was there still a response to Zaprinast after 48 h knockdown, where the thapsigargin response was apparently fully ablated?

Unfortunately, we were unable to perform this experiment as it is not possible to obtain sufficient cells at 48 h with ATc. Due to the essential role of TgSERCA, parasites are unable to replicate after 24 h.

As noted in the public comments, the data in Figure 4A vs 4G and Figure 4B vs 4H appear to show that the calcium responses to GPN are similar to that with thapsigargin, which seems unexpected if the acidic compartment is loaded from the ER. The results with GPN addition after thapsigargin (Figure 1H) argue against this, but the authors should still cite the work of Atakpa et al.

We think that the reviewer is concerned that GPN may also be acting on the ER. This is a possibility that we considered, and we now included the suggested citation (line 457). However, we believe that it is difficult to directly compare the responses, as the kinetics of calcium release from the ER may differ from those of release from the PLVAC. This could be due to differences in the calcium buffering capacity between the two compartments. Additionally, it is possible that calcium leaked from the ER is more efficiently sequestered by other stores or extruded through the plasma membrane than calcium released from the PLVAC. Besides, GPN is known to have a more disruptive effect on membranes compared to TG, which may also influence their responses. As noted by the reviewer, Figure 1H also supports the idea that the acidic compartment is loaded from the ER.

The abbreviation for the plant-like vacuolar compartment (PLVAC) only appears in a figure legend but should be defined in the main text on first use.

Corrected, lanes 140-143

The authors should cite the previous study of Borges-Pereira et al., 2020 (PMID: 32848018) that also demonstrates the incomplete overlap of the calcium pools mobilized by thapsigargin and CPA in *P. falciparum*. The ability to measure calcium in intracellular stores using MagFluo4 opens the possibility to further investigate this discrepancy between CPA and thapsigargin, but CPA does not appear to have been used in the permeabilized cell experiments with MagFluo4. I would suggest that this could be added to Figure 2 and/or Figure 4, or at least as a supplementary figure.

In response to this reviewer’s critique we performed additional experiments with Mag-Fluo4 loaded parasites. These are presented in the new Figure S3. We added CPA and TG and combined them to inhibit SERCA and to allow calcium leak from the loaded organelle. Under these conditions, we observed a very similar leak rate after the addition of the inhibitors as measured by the slope of Ca^2+^ leak. We believe that the leak rate is most likely determined by the intrinsic ER mechanism. See the discussion of this result in lines 436442 and the previous response to the same reviewer comment.

**Reviewer #3 (Recommendations for the authors):**
Suggestions for improved or additional experiments, data, or analyses(1) Figure 1A is not mentioned in the main text even though it is discussed.

Corrected.

(2) Figure 1G: Values do not match, how can GPN be so high?

These figures were replaced by new traces and individual quantification analyses for each experiment.

(3) Figure 1H and I: Is this type of data/results also available for the mitochondrion?

Unfortunately, we were not able to include this experiment because we were unable to accurately quantify the mitochondrial calcium release. Instead, we used mitochondrial GECIs and the results are shown in Figure 5 to study mitochondrial calcium uptake.

(4) Figure 1H: where does the calcium go after GPN addition? Taken up by another calcium store?

Most likely calcium is extruded through the plasma membrane by the activity of the Calcium ATPase TgA1.

However, the reviewer’s suggestion is also possible, and calcium could be taken by another store like the mitochondrion. In this regard, we did observe a large mitochondrial calcium increase (parasites expressing SOD2-GCaMp6) after adding GPN (Fig 5I) suggesting that the mitochondrion may take calcium from the organelle targeted by GPN. However, the calcium affinity of the mitochondrion is very low, so the concentration of calcium needs to be very high to activate it and these concentrations are most likely achieved at the microdomains formed between the mitochondrion and other organelles.

(5) Figure 2B-C: Further explanation of why these particular values were chosen for the follow-up experiments would be helpful for the reader.

We tested a wide range of MgATP and free calcium concentrations to measure ER Ca^2+^ uptake catalyzed by TgSERCA. The concentrations shown fall within the linear range.

We followed the free calcium concentrations used by studies of mammalian SERCA (https://doi.org/10.1016/j.ceca.2020.102188). In this protocol they used 220 nM free calcium, which was close to cytosolic Ca^2+^ levels. TgSERCA can take up calcium efficiently at this concentration, as shown in Fig 2. We used less MgATP than the mammalian cell protocols, since we did not observe a significant increase in SERCA activity beyond 0.5 mM MgATP. We added one more sentence explaining in the results, lines 204-207.

(6) Figure 3E: Revise the error bar? (and note that colours do not match the graph legend).

The colors do match; the problem visualizing it is because vacuoles containing a single parasite are virtually absent in the control group without ATc treatment.

(7) Figure 3H: 'Interestingly, when testing egress after the addition of saponin in the presence of extracellular Ca2+, we observed that the tachyzoites egressed sooner (Figure 3H, saponin egress).' This is the only graph showing egress timing, and thus it is not clear what is the comparison. The egressed here is sooner compared to what condition? Egress in the absence of Ca2+? This requires clarification and might require the control data to be added.

In the saponin experiment we compare time to egress of the mutant grown with or without ATc. The measurement is for time to egress after adding saponin. This experiment is in the presence of extracellular calcium. The protocol was previously used to measure time to egress: PMID: 40043955, PMID: 38382669, PMID: 26374900. See also response to question 9 of reviewer 1.

(8) Figure 4C: There is a small peak appearing right after TG addition this should be discussed and explained.

This trace was generated in a different fluorometer, F-4000. This was an artifact due to jumping of the signal when adding TG. Multiple repeats of the same experiment in the newer F7000 did not show the peak. We included in the MM the use of the F-4000 fluorometer for some experiments. We apologize for the omission. Lines 609-610

(9) Figure 5A: An important control that is missing is co-localisation with a mitochondrial marker.

The expression of the SOD2-GCaMP6 has been characterized: PMID: 31758454

(10) Figure 5H: This line was made for this study however the line genetic verification is missing.

In response to this concern we now include a new Figure S5 showing the fluorescence of GCaMP6 in the mitochondrion of the iDTgSERCA mutant (Fig. S5A). We include several parasites. In addition, we show fluorescence measurements after addition of Calcium showing that the cells are unresponsive indicating that the indicator is not in the cytosol. Lines 650-651 and 344-348.

(11) Figure 6D: since the membranes are hard to see, it is not clear whether the arrows show structures that are in line with the definition of membrane contact sites. The authors should provide an in-depth analysis of the length of the interaction between the membranes where the distance is less than 30 nM, and discuss how many structures corresponding to the definition were analysed.

All the requested details are now included in the legend to Figure S3.

Minor corrections to the text and figures(1) Unify statistical labelling throughout the paper replacing *** with p values.

Corrected. We changed the *** with the actual p value in some figures. For figure 2 and Fig S1, we still use the *** due to the space limitation.

(2) Unify ATC vs ATc throughout the paper.

Corrected.

(3) Unify capitalization of line name (iΔTgserca/i ΔTgSERCA) throughout the paper.

Corrected.

(4) Unify capitalization of p value (p/P) throughout the paper.

Corrected in figures.

(5) Unify Fig X vs Fig. X throughout the text.

Corrected.

(6) Add values of scale bars to legends (eg Figure S2).

Corrected.

(7) What is the time point for the data in Figures 4E-H, 5H, and S3? 24hrs? include in the legend.

Added 24 h to the legends. Fig S3 is now S4.

(8) Figure 3F: The second graph is NS thus perhaps no need for the p-value?

Corrected.

(8) Figure 3G: Worth considering swapping the two around: first attachment and then invasion?

Corrected. Invasion and attachment bars were swapped.

(10) Figure 4A/B: Wrong colour match for Figure 4B.

Corrected.

(11) Figure 4F: In the main text, the authors reference to Figure 1F, correct to 4F.

Corrected

(12) Figure 4H: In the main text, authors reference to Figure 1H, correct to 4H.

Corrected.